# Designing a Structural Health Monitoring System Accounting for Temperature Compensation

**Valeria Francesca Caspani, Daniel Tonelli *, Francesca Poli and Daniele Zonta**

Department of Civil Environmental and Mechanical Engineering, University of Trento, 38123 Trento, Italy; valeria.caspani@unitn.it (V.F.C.); francesca.poli-2@unitn.it (F.P.); daniele.zonta@unitn.it (D.Z.)

*   Correspondence: daniel.tonelli@unitn.it

**Abstract:** Structural health monitoring is effective if it allows us to identify the condition state of a structure with an appropriate level of confidence. The estimation of the uncertainty of the condition state is relatively straightforward a posteriori, i.e., when monitoring data are available. However, monitoring observations are not available when designing a monitoring system; therefore, the expected uncertainty must be estimated beforehand. This paper proposes a framework to evaluate the effectiveness of a monitoring system accounting for temperature compensation. This method is applied to the design process of a structural health monitoring system for civil infrastructure. In particular, the focus is on the condition-state parameters representing the structural long-term response trend, e.g., due to creep and shrinkage effects, and the tension losses in prestressed concrete bridges. The result is a simple-to-use equation that estimates the expected uncertainty of a long-term response trend of temperature-compensated response measurements in the design phase. The equation shows that the condition-state uncertainty is affected by the measurement and model uncertainties, the start date and duration of the monitoring activity, and the sampling frequency. We validated our approach on a real-life case study: the Colle Isarco viaduct. We verified whether the pre-posterior estimation of expected uncertainty, performed with the experimented approach, is consistent with the real uncertainty estimated a posteriori based on the monitoring data.

**Keywords:** structural health monitoring; temperature compensation; monitoring system design; monitoring capacity; pre-posterior analysis; long-term structural response; parameter estimation; uncertainty quantification; Bayesian data analysis; prestressed concrete bridge

## 1. Introduction

When monitoring a civil structure, the purpose is to identify a number of key parameters—including modal frequencies, long-term strain trends and damage indicators [1,2]—which are representative of its health state. The monitoring is successful if these parameters can be identified with an appropriate level of confidence, keeping their uncertainty below an acceptable target value [3]. Acceptable uncertainty normally depends on the nature of the physical problem that is being investigated and could be required by the user of the monitoring information within the management framework of the infrastructure [4,5]. In any case, controlling parameter uncertainty is as important as the identification of the parameter values themselves.

Estimating the uncertainty of key parameters is relatively straightforward a posteriori, i.e., when monitoring data are available. Data analysis methods for the estimation of the posterior uncertainty of parameters range from plain least squares [6] to more sophisticated Markov-chain Monte Carlo sampling [7,8] and to other probabilistic machine learning techniques [9]. However, monitoring observations are not yet available when designing a monitoring system; therefore, in order to understand whether a monitoring strategy (the choice of sensors, sensor placement and acquisition protocols) is suitable, it is necessary to estimate an expected uncertainty of the key parameters beforehand. This expected

uncertainty is indeed an a priori estimation (i.e., before acquiring any data) of posterior uncertainty: it is sometimes referred to, in short, as pre-posterior uncertainty, and the estimation process of this value is known as pre-posterior analysis [3].

The optimal sensor placement is a typical instance of bridge monitoring problems solved through pre-posterior analysis: the problem is the choice of the optimal configuration of a fixed number of sensors to maximize the acquired information. Papadimitriou et al. [10] proposed a method based on the minimization of information entropy, defined as the parameter that quantifies the uncertainty of a random variable; Udwadia [11] proposed a method based on the maximization of the Fisher information matrix norm; Fedorov and Hackl proposed the individuation of the optimal sensor configuration in the one that minimizes the coefficients of the variance and covariance matrix [12]. In all of these problems, the choice of the configuration and technical specifications of sensors is based on the minimization of a value representing the expected uncertainty of the key parameters.

This paper focuses on a different typology of bridge monitoring problems, where the key parameters to be identified are long-term response trends, e.g., strain ($\mu\varepsilon$/year), deflection (mm/year) or rotation (mrad/year) trends. Long-term response drifts are very useful to predict the future behaviour of a structure, and to understand ahead of time whether the bridge may transit towards a damaged state (e.g., cracking) or exceed the value for the serviceability limit (e.g., excessive deflection). Sudden changes in the response trends are also typical symptoms of the occurrence of damage (e.g., the failure of one or more prestressing tendons) [13].

Long-term deflection trends are particularly evident in prestressed concrete bridges [14–16]. Hubler et al. [17] analysed the vertical deflection of 63 bridges with different structural types, ages, and deterioration states. They pointed out that all of the bridges experienced a long-term deflection trend and exceeded the design predictions. They also developed a predictive model of creep and shrinkage, named Model B3 [18], for a more realistic assessment of long-term concrete phenomena, which significantly affect the durability and life-long functionality of bridges. This model better predicts the experimental evidence than older ones commonly used in bridge design, such as ACI 209 [19], CEB [20], and EN1992 [21]. In particular, it clearly explains the excessive deformation and deflection of prestressed reinforced concrete bridges with box girder sections (e. g., Colle Isarco Viaduct [22] and Koror-Babeldaob Bridge [23]). Another frequent parameter is the trend in the loss of prestressing force [24].

Usually, the long-term response trends of bridges have a small entity, and their typical values are lower than 30 $\mu\varepsilon$/year for strains, 0.50 mm/year for deflections, and 10 $\mu$rad/year for rotations [17,25]. Hence, these trends are difficult to recognize in a data record that is strongly affected by traffic loads and temperature variation. The use of indirect measurement produces additional uncertainty and noise. For example, the loss of prestress in the tendons is commonly estimated indirectly using strain sensors embedded in the concrete [25,26], as direct measurement methods (e.g., using load cells) are often unfeasible.

Among the different sources of uncertainty, temperature is usually the most important by far [27]. Indeed, besides the complex phenomena that can occur at elevated temperatures [28,29], daily and seasonal temperature variations also have a relevant effect on structural responses. For example, the value of the permanent contraction drift in a concrete member due to creep or shrinkage typically ranges around a few $\mu\varepsilon$ per year, while the sole daily thermal variation is in the order of 100–200 $\mu\varepsilon$ per day [30]. Extracting that drift requires us to perform a thermal compensation on a signal dominated by the thermal effect, and this will inevitably introduce some errors that must be controlled. When data are available, thermal compensation can be performed by fitting monitoring data with a model that considers the effects of temperature [31]. By fitting the monitoring data with a probabilistic method [6], the posterior uncertainty introduced by the thermal compensation can be quantified as well, in order to evaluate whether its magnitude is acceptable or not.

However, at the design stage, no monitoring system is installed and no recording is available. As such, how is it possible to predict the error introduced by the thermal compensation? How can we design a proper monitoring system to keep the uncertainty below the limit values?

This paper aims to answer these questions by introducing a logical approach for a pre-posterior estimation of the expected uncertainty of a long-term structural response trend. The proposed formulation considers the errors related to temperature compensation, sensor performance, and the interpretation model. It allows the estimation of the expected uncertainty before monitoring data are available, thus helping monitoring system designers to answer the following design questions: (i) What sensor technology and measurement accuracy are required? (ii) What should be the duration of the monitoring, in order to verify the target uncertainty? (iii) What is the minimum sampling frequency to satisfy the target uncertainty, given a certain monitoring duration?

This paper is organized as follows. Section 2 outlines the underlying assumptions of the approach, and the development of the formulation for the evaluation of the uncertainty of the trend parameters. Section 3 contains the analysis of the variation of the expected uncertainty of the trend parameters according to the sensor performance, model uncertainty, sampling frequency, monitoring duration and start date of the monitoring acquisition. Section 4 introduces the Colle Isarco Viaduct case study [32] and describes the monitoring system that is currently installed on it. The Colle Isarco Viaduct is one of the longest statically determinate, prestressed concrete highway bridges in the European Alpine region. An abnormal progressive deflection of the main span has been observed on it, due to a combination of creep effects and the loss in prestressing force; recently, a drastic retrofit intervention was performed on the structure [22]. In Section 5, the approach for the estimation of the pre-posterior uncertainty of the long-term local-strain trend is applied on the concrete box girders of the Colle Isarco Viaduct. Furthermore, a pre-posterior calculation is performed by the analysis of different approaches: (1) in the first, the temperature is modelled as a sine function; (2) in the second, the temperature is modelled as a sine function with the addition of Gaussian noise. The uncertainty of the trend parameter is calculated a posteriori based on the monitoring data, then the pre-posterior and posterior results are compared in order to validate the approach used. Finally, in Section 6, some conclusions are drawn.

## 2. Problem Statement and Formulation

Assume that we are monitoring the response $y$ (e.g., displacement, deflection, rotation, stress, strain, modal frequency) of a structural element, and that this response is strongly affected by temperature $T$. Assume that the monitoring consists of a total number of $N$ samples, and indicate with $y_i$ and $T_i$ the response and temperature sample simultaneously recorded at time $t_i$, with index $i = 1, \ldots ,N$. Let us indicate with $N \times 1$ vectors **y**, **T**, and **t**, respectively. Assume that the measurements are independent, and that a daily thermal compensation has been performed, while the measurements are not compensated for the seasonal thermal variations.

It is often convenient to express time and temperature in relative terms: label $\Delta t = t_i - t_0$ and $\Delta T = T_i - T_0$ where $t_0$ and $T_0$ are an arbitrary reference time and temperature. For instance, $t_0$ could be the starting time of the monitoring, and $T_0$ could be equal to $0\,^{\circ}\text{C}$.

Let $m$ be the permanent trend of variation over the time of the response $y(t)$ (e.g., the deflection trend of a cantilever bridge, the local contraction or expansion trend of concrete), purged from the seasonal temperature effect. The goal is to identify the most probable value of $m$ and its uncertainty $\sigma$, given the monitoring observations. For this purpose, we define a probabilistic interpretation model, stating the relationship between measurements $y_i$, temperature $T_i$ and time $t_i$.

1.  Assume a linear interpretation model, in the form:

$$y_i = y_0 + m \cdot \Delta t_i + \alpha \cdot \Delta T_i + z_i \qquad (1)$$

where:

- $y_0$ is an offset parameter representing the ideal strain at time $t_0$ and temperature $T_0$.
- $\alpha$ is the apparent thermal expansion coefficient. Because the thermal expansion coefficients of common building materials do not show significant variations within the range of temperatures where they are designed to be used [21], $\alpha$ is assumed to be a constant value regardless of the temperature changes.
- $m$ is the variation trend over time, which is the object of the estimation. It is the gradient of the linear model with respect to time. It expresses the linear trend over time of $y(t)$ purged from temperature effects, and includes all long-term effects. We model the long-term effects with a linear trend because we consider the measurements acquired during a relatively short time interval (one–two years of monitoring) [7].
- $z_i$ is the residual, the difference between the statistically independent observation $y_i$ and the nominal value of the model, $y_0 + m \cdot \Delta t_i + \alpha \cdot \Delta T_i$.
- In practice, the interpretation model that connects the temperature, time, and response measurements is controlled by three unknown parameters, clustered into the vector $\theta$ = $\{y_0, m, \alpha\}$.
- Let us label **D** the Jacobian matrix of the interpretation model with respect to the parameters:

$$\mathbf{D} = \begin{bmatrix} \frac{\partial y}{\partial y_0} & \frac{\partial y}{\partial m} & \frac{\partial y}{\partial \alpha} \end{bmatrix} = \begin{bmatrix} 1 & \Delta t_1 & \Delta T_1 \\ \vdots & \vdots & \vdots \\ 1 & \Delta t_N & \Delta T_N \end{bmatrix} = \begin{bmatrix} 1 & \mathbf{\Delta t} & \mathbf{\Delta T} \end{bmatrix} \tag{2}$$

As a result, Equation (1) can be written in a matrix form:

$$\mathbf{y} = \mathbf{D} \cdot \theta + \mathbf{z} \tag{3}$$

2. A priori model parameters are Gaussian independent random variables with a mean value $\mathbf{\mu_\theta}$ and covariance $\mathbf{\Sigma_\theta}$ equal to

$$\mathbf{\Sigma_\theta} = \begin{bmatrix} \sigma_{y0}^2 & 0 & 0 \\ 0 & \sigma_\alpha^2 & 0 \\ 0 & 0 & \sigma_m^2 \end{bmatrix} \tag{4}$$

3. Residuals vector **z** is distributed as zero-mean Gaussian noise with known variance $\sigma_{\text{LH}}{}^2$.

The goal is to estimate the most probable value of parameter $m$ and its uncertainty $\sigma_{m|\mathbf{y}}$ based on monitoring observation **y**; to do so, the prior distribution of the model parameter p($\theta$) is updated into the posterior distribution p($\theta$ | **y**) through Bayes' rule [33]:

$$p(\theta \,|\, \mathbf{y}) = p(\mathbf{y} \,|\, \theta) \cdot p(\theta) / p(\mathbf{y}) \tag{5}$$

where p(**y** | $\theta$) is the likelihood function, i.e., the probability of observing measurement **y** given the interpretation model and the parameters vector $\theta$; p(**y**) is a normalization constant referred to as evidence.

According to Hp. 2 and Hp. 3 (linear Gaussian model), the joint probability distribution p(**y** | $\theta$)· p($\theta$) over parameters vector $\theta$ is Gaussian [33]. The aim is to estimate the mean values of the model parameters that maximize the posterior probability distribution p($\theta$ | **y**), $\theta_{\text{MAP}}$, and their covariance matrix $\mathbf{\Sigma_{\theta|y}}$.

$$\theta_{\text{MAP}} = \underset{\theta}{\text{argmax}}\, p(\theta|\mathbf{y}) = \underset{\theta}{\text{argmax}}\{p(\mathbf{y}|\theta) \cdot p(\theta)\} \tag{6}$$

In practice, it is more convenient to maximize the log of a Gaussian posterior distribution p($\theta$ | **y**) than the Gaussian distribution itself. Indeed, the logarithm is a monotonically

increasing function of its argument, and the maximization of the log of a function is equivalent to the maximization of the function itself [33]. As a result, Equation (6) becomes:

$$\theta_{MAP} = \underset{\theta}{\mathrm{argmax}}\{\log\ p(\mathbf{y}|\theta) + \log\ p(\theta)\} = \underset{\theta}{\mathrm{argmin}}\left\{\frac{1}{2\cdot\sigma_{LH}^2}\mathbf{z}^T\mathbf{z} + \frac{1}{2}(\theta - \mu_\theta)^T\Sigma_\theta^{-1}(\theta - \mu_\theta)\right\} \tag{7}$$

The value of $\theta$ that maximizes $p(\theta\,|\,\mathbf{y})$ can be calculated by setting as equal to zero the derivative of the log-posterior with respect to $\theta$, while the inverse of the a posteriori covariance matrix can be calculated as the second-order derivative of the log-posterior with respect to $\theta$. As a result, by combining Equations (3) and (7), we arrive at

$$\begin{cases} \theta_{\mathbf{MAP}} = \mu_\theta + \frac{1}{\sigma_{LH}^2}\Sigma_{\theta|\mathbf{y}}\mathbf{D}^T(\mathbf{y} - \mathbf{D}\cdot\mu_\theta) \\ \Sigma_{\theta|\mathbf{y}}^{-1} = \frac{1}{\sigma_{LH}^2}\mathbf{D}^T\mathbf{D} + \Sigma_\theta^{-1} \end{cases} \tag{8}$$

We can explicitly compute matrix $\Sigma_{\theta\,|\,\mathbf{y}}{}^{-1}$ in Equation (8) with the matrices $\mathbf{D}$ and $\Sigma_\theta$ defined in Equations (2) and (4):

$$\Sigma_{\theta|\mathbf{y}}^{-1} = \frac{1}{\sigma_{LH}^2}\begin{bmatrix} \mathbf{1}^T\mathbf{1} & \mathbf{1}^T\Delta\mathbf{T} & \mathbf{1}^T\Delta\mathbf{t} \\ \mathbf{1}^T\Delta\mathbf{T} & |\Delta\mathbf{T}|^2 & \Delta\mathbf{T}^T\Delta\mathbf{t} \\ \mathbf{1}^T\Delta\mathbf{T} & \Delta\mathbf{T}^T\Delta\mathbf{t} & |\Delta\mathbf{t}|^2 \end{bmatrix} + \begin{bmatrix} \frac{1}{\sigma_{y_0}^2} & 0 & 0 \\ 0 & \frac{1}{\sigma_\alpha^2} & 0 \\ 0 & 0 & \frac{1}{\sigma_m^2} \end{bmatrix} \tag{9}$$

where $|\Delta\mathbf{v}| = \Delta\mathbf{v}^T\Delta\mathbf{v}$ indicates the Eulerian norm of the generic vector v. It is convenient to express the relative time and temperature $\Delta\mathbf{t}$ and $\Delta\mathbf{T}$ with respect to the mean values $\overline{\mathbf{t}} = \mathbf{1}^T\Delta\mathbf{t}/N$ and $\overline{\mathbf{T}} = \mathbf{1}^T\Delta\mathbf{T}/N$ of the vectors time $\mathbf{t}$ and temperature $\mathbf{T}$:

$$\begin{cases} \Delta\mathbf{t} = \mathbf{t} - \overline{\mathbf{t}} \\ \Delta\mathbf{T} = \mathbf{T} - \overline{\mathbf{T}} \end{cases} \tag{10}$$

As a result, Equation (9) becomes

$$\Sigma_{\theta|\mathbf{y}}^{-1} = \mathbf{D}^T\Sigma_{\mathbf{y}|\theta}^{-1}\mathbf{D} + \Sigma_\theta^{-1} = \frac{1}{\sigma_{LH}^2}\begin{bmatrix} N & 0 & 0 \\ 0 & |\Delta\mathbf{T}|^2 & \Delta\mathbf{T}^T\Delta\mathbf{t} \\ 0 & \Delta\mathbf{T}^T\Delta\mathbf{t} & |\Delta\mathbf{t}|^2 \end{bmatrix} + \begin{bmatrix} \frac{1}{\sigma_{y_0}^2} & 0 & 0 \\ 0 & \frac{1}{\sigma_\alpha^2} & 0 \\ 0 & 0 & \frac{1}{\sigma_m^2} \end{bmatrix} \tag{11}$$

The posterior uncertainty $\sigma_{m\,|\,\mathbf{y}}$ of the trend-parameter $m$ is the third diagonal element of the posterior covariance matrix $\Sigma_{\theta\,|\,\mathbf{y}}$. After a simple mathematical manipulation of Equation (11), the posterior uncertainty can be expressed as:

$$\sigma_{m|\mathbf{y}}(\mathbf{T},\mathbf{t}) = \sigma_{LH}\cdot\frac{1}{\sqrt{\frac{\sigma_{LH}^2}{\sigma_m^2} + |\Delta\mathbf{t}|^2}}\cdot\frac{1}{\sqrt{1 - \left(\frac{|\Delta\mathbf{t}|^2}{\frac{\sigma_{LH}^2}{\sigma_m^2}+|\Delta\mathbf{t}|^2}\right)\left(\frac{|\Delta\mathbf{T}|^2}{\frac{\sigma_{LH}^2}{\sigma_\alpha^2}+|\Delta\mathbf{T}|^2}\right)\rho_{\mathbf{tT}}^2}} \tag{12}$$

where $\rho_{\mathbf{tT}}$ is the Pearson correlation coefficient [34] between the time and temperature, i.e., the ratio between the covariance of the two variables and the product of their standard deviations:

$$\rho_{\mathbf{tT}} = \frac{\Delta\mathbf{t}\cdot\Delta\mathbf{T}}{\sqrt{|\Delta\mathbf{t}|^2|\Delta\mathbf{T}|^2}} = \frac{\sum_{i=1}^N(t_i - \overline{\mathbf{t}})(T_i - \overline{\mathbf{T}})}{\sqrt{\sum_{i=1}^N(t_i - \overline{\mathbf{t}})^2}\sqrt{\sum_{i=1}^N(T_i - \overline{\mathbf{T}})^2}} \tag{13}$$

The Pearson correlation coefficient ranges between $-1$ and $+1$, with $+1$ meaning perfect direct correlation, $-1$ being perfect inverse correlation, and $0$ being independency between the two data records. In this specific case, $\rho_{\mathbf{tT}}$ is close to 1 if the temperature history can be approximated to a straight line, and it is close to 0 if the temperature and time are far from a linear relationship.

Equation (12) provides an explicit expression of the posterior uncertainty of a linear trend $m$, which depends on the prior uncertainty $\sigma_m^2$ of $m$, monitoring noise, the extension of time sampling $|\mathbf{\Delta t}|$, and the correlation between temperature and time $\rho_{\mathbf{tT}}$, although the dependence on these quantities is not exactly intuitive.

When the prior parameters are highly uncertain ($\sigma_m \to \infty$ and $\sigma_\alpha \to \infty$), or the prior information is negligible, Equation (12) is particularly simple and easy to read:

$$\sigma_{m|\mathbf{y}}(\mathbf{T},\mathbf{t}) = \sigma_{\mathrm{LH}} \cdot \frac{1}{|\mathbf{\Delta t}|} \cdot \frac{1}{\sqrt{1 - \rho_{\mathbf{tT}}^2}} \tag{14}$$

This equation clearly shows that the posterior uncertainty of parameter $m$ is the combination of three different factors:

- The monitoring noise $\sigma_{\mathrm{LH}}$, accounting for the measurement of the noise and the uncertainty of the hypothesized linear model.
- The term $1/|\mathbf{\Delta t}|$, which effectively depends on the monitoring duration and the sampling rate.
- The term $1/\sqrt{1 - \rho_{\mathbf{tT}}^2}$, which depends on the extent to which the temperature history is close to a straight line.

## 3. Application to Monitoring System Design

Equations (12) and (14) allow the calculation of the uncertainty $\sigma_{m|\mathbf{y}}$ of a linear fit a posteriori, i.e., after monitoring data are acquired. In the design stage of the monitoring system, the goal is to predict the uncertainty of $m$ that is expected in the monitoring, even if no data are currently available. Let $\sigma_{m,\mathrm{pp}}$ be the a priori estimate (i.e., before data is acquired) of the posterior uncertainty of parameter $m$. This quantity is also referred to as pre-posterior uncertainty (hence the "pp" in the symbol), in order to distinguish it both from the prior uncertainty $\sigma_m$ (the uncertainty of the parameter if no monitoring is carried out) and the posterior uncertainty $\sigma_{m|\mathbf{y}}$ (the uncertainty after monitoring data are acquired).

A notable feature of both Equations (12) and (14) is that the posterior uncertainty is completely independent from the response recording $\mathbf{y}$. Therefore, it is possible to estimate the pre-posterior uncertainty $\sigma_{m,\mathrm{pp}}$ by making reasonable assumptions on the time sampling vector $\mathbf{t}$, the expected temperature record $\mathbf{T}$, and the value of the residual's noise $\sigma_{\mathrm{LH}}$.

### 3.1. Pre-Posterior Estimate of Time and Temperature Vectors

Assuming that the monitoring sampling is uniform, the time vector only depends on the total monitoring time $t_{tot}$ and the design sampling frequency $f_s$. The total number of measurements $N$ acquired during the monitoring is

$$N = t_{tot} \cdot f_s + 1 \tag{15}$$

and the timestamps of the measurements are defined as

$$t_i = \frac{1}{f_s}(i - 1), \ i = 1, 2, 3, \ldots, N \tag{16}$$

For design purposes, the expected temperature measurements $T_i$ can be expressed through a sine function with a period $\tau = 365.25$ days, or 1 year [24]:

$$T_i = a \cdot \sin\left(\frac{2\pi}{\tau} t_i + b\right) + c, \quad i = 1, 2, 3, \ldots, N \tag{17}$$

where $a$ is the amplitude of the temperature sinusoidal function, $b$ is the phase, and $c$ is an offset corresponding to the temperature's seasonal mean value.

### 3.2. Pre-Posterior Estimate of the Residual Noise $\Sigma_{lh}$

Assume that sensor measurements $y_i$ and $T_i$ are approximations of the corresponding true physical values $\hat{y}(t_i)$ and $\hat{T}(t_i)$, and that the scatter between the two is zero-mean random noise $n_j(\sigma^2)$. Let the variances of the measurements be $\sigma_y{}^2$ and $\sigma_T{}^2$, depending on sensor accuracy:

$$\begin{cases} y_i = \hat{y}(t_i) + n_i \sigma_y{}^2 \\ \Delta T_i = \Delta \hat{T}(t_i) + n_i \sigma_T{}^2 \end{cases} \tag{18}$$

Given the linear interpretation model defined, we assume that the true values satisfy

$$\hat{y}(t_i) = y_0 + m \cdot \Delta t_i + \alpha \cdot \Delta \hat{T}(t_i) + n_i\left(\sigma^2_{model}\right) \tag{19}$$

where $n_i(\sigma^2{}_{model})$ is zero-mean normal noise with variance $\sigma^2{}_{model}$ that represents the difference between the true physical quantity and the model prediction. The model uncertainty $\sigma^2{}_{model}$ derives from approximations and idealizations made in the formulation of the interpretation model, as well as in the choice of the probability distribution of the model parameters.

Then, Equations (18) and (19) can be merged into a single equation:

$$y_i - n_i(\sigma_y^2) = y_0 + m \cdot t_i + \alpha \cdot (T_i - n_i(\sigma_T^2)) + n_i(\sigma^2_{model}) \tag{20}$$

and all of the zero-mean Gaussian errors can be merged into one zero-mean Gaussian error $n_i(\sigma_{LH}{}^2)$, which corresponds precisely to the residual $z_i$ between the observation $y_i$ and the nominal value of the model:

$$n_i(\sigma^2_{LH}) = z_i = n_i(\sigma_y^2) + n_i(\sigma^2_{model}) + \alpha \cdot n_i(\sigma_T^2) \tag{21}$$

As a result, variance $\sigma_{LH}{}^2$ includes both the noise of the sensors and the uncertainty of the model, combined with the propagation of uncertainty through the square root of the sum of squares, assuming statistically uncorrelated errors [34]:

$$\sigma_{LH}{}^2 = \sigma_y^2 + \alpha^2 \cdot \sigma_T^2 + \sigma^2_{model} \tag{22}$$

### 3.3. Impact of the Time–Temperature Correlation

The term $1/\sqrt{(1 - \rho_{tT}^2)}$ in Equation (14) depends on the linear correlation coefficient $\rho_{Tt}$ between the time and temperature vectors. Given the definition of $T_i(a,b,c)$ in Equation (17), the coefficient $\rho_{Tt}$, in turn, depends on the phase-parameter $b$. In practice, this parameter allows the setting of the start date of the monitoring period. On the other hand, both the parameters amplitude $a$ and offset $c$ do not have any influence on $\rho_{Tt}$; therefore, they can be set arbitrarily.

Given two sine temperature functions like Equation (17), if their differences with regard to phase-parameter $b$ are equal to $\pi$, e.g., $\mathbf{T}(b = 0)$ and $\mathbf{T}(b = \pi)$, the results in terms of $\sigma_{m,pp}$ are the same. This is due to the presence of the square value of $\rho_{Tt}$ in $1/\sqrt{(1 - \rho_{tT}^2)}$,

which makes it irrelevant whether the time and temperature vectors are directly or inversely correlated:

$$\sigma_{m,\text{pp}}(\mathbf{T}_{b=0}) = \sigma_{m,\text{pp}}\left(\mathbf{T}_{b=\pi}\right) \tag{23}$$

In contrast, if the difference in the phase-parameter $b$ is equal to $\pi/2$, e.g., $\mathbf{T}(b = 0)$ and $\mathbf{T}(b = \pi/2)$, the result is the maximum difference $\Delta_{\max}$ between the values of $\sigma_{m,\text{pp}}$ estimated with those two temperature vectors:

$$\sigma_{m,\text{pp}}(\mathbf{T}_{b=0}) - \sigma_{m,\text{pp}}(\mathbf{T}_{b=\pi/2}) = \Delta_{\max} \tag{24}$$

Because the sinusoidal function $T$ has a period of $\tau = 365$ days, a difference in $b$ equal to $\pi$ corresponds approximately to 6 months ($\tau/2 = 182.5$ days $\sim$ 6 months); in contrast, a difference in $b$ equal to $\pi/2$ approximately corresponds to 3 months ($\tau/4 \sim 91.25$ days $\sim$3 months).

Figure 1 shows the impact of the time–temperature correlation on $\sigma_{m,\text{pp}}$; specifically, it shows the comparison between the two limit cases, $b = 0$ and $b = \pi/2$, in terms of $\rho_{\mathbf{Tt}}$, $1/\sqrt{(1 - \rho_{\mathbf{tT}}^2)}$, and $\sigma_{m,\text{pp}}$.

Figure 1b shows the absolute value of $\rho_{\mathbf{Tt}}$ as the monitoring duration increases, while Figure 1c,d show the effect of $|\rho_{\mathbf{Tt}}|$ on $1/\sqrt{(1 - \rho_{\mathbf{tT}}^2)}$ and $\sigma_{m,\text{pp}}$; different values of the phase parameter $b$ in the temperature function $T(b)$ determine different outcomes.

When $b = 0$ (red curves), the absolute value of the correlation coefficient $\rho_{\mathbf{Tt}}$ decreases to 0 after approximately 183 days from the monitoring start date. Then, it increases until 320 days, when it decreases again; it reaches 0 for the second time after around 540 days from the monitoring start date. In contrast, when $b = \pi/2$ (blue curves), $|\rho_{\mathbf{Tt}}|$ increases until 180 days from the monitoring start date. Then, from 180 to 360 days, $|\rho_{\mathbf{Tt}}|$ decreases monotonically. It is interesting to highlight that for longer monitoring durations, the linear correlation between the time and temperature increases and decreases periodically, without zeroing permanently. However, such variations in $|\rho_{\mathbf{Tt}}|$ do not have a great influence on $1/\sqrt{(1 - \rho_{\mathbf{tT}}^2)}$, or consequently on $\sigma_{m,\text{pp}}$. Indeed, after approximately 450 days of monitoring, $5\tau/4 \sim 457$ days $\sim$15 months, the term $1/\sqrt{(1 - \rho_{\mathbf{tT}}^2)}$ is approximately equal to 1. Therefore, the impact of the time–temperature correlation on $\sigma_{m,\text{pp}}$ is negligible for long monitoring periods (more than 15 months).

Figure 1d shows that the difference between the two limit cases $\sigma_{m,\text{pp}}(\mathbf{T}_{b=0})$ and $\sigma_{m,\text{pp}}(\mathbf{T}_{b=\pi/2})$ also becomes negligible after around 15 months of monitoring; therefore, the start date of the monitoring does not influence the monitoring effectiveness.

From Figure 1d, it can also be noted that the red curve $\sigma_{m,\text{pp}}(\mathbf{T}_{b=0})/\sigma_{\text{LH}}$ has a constant plateau from day 183 to day 320; the expected uncertainty remains constant without decreasing for a long interval of the monitoring period due to the increasing linear correlation $\rho_{\mathbf{Tt}}^2$ during such an interval. The practical meaning of this observation is that increases in the monitoring period within this interval will not produce an improved knowledge of the structural state in terms of the measurement trend, due to the effect of temperature.

It may be observed that a theoretical sinusoidal function like $T(t_i,a,b,c)$ in Equation (17) might not be representative of experimental temperature measurements. Real temperature measurements can be better simulated by adding Gaussian noise $n_i(\sigma_{noise})$ to the sinusoidal temperature, which represents the observed variation in real temperature between one day and the following one:

$$T_{i,noise} = T_i + n_i(\sigma_{noise}) \tag{25}$$

where $\sigma_{noise}$ can be set, for instance, as 5% of the temperature range of the sine function. Figure 2 shows the impact of temperature variation simulated as Gaussian noise $n_i(\sigma_{noise})$ on the pre-posterior uncertainty $\sigma_{m,\text{pp}}$.

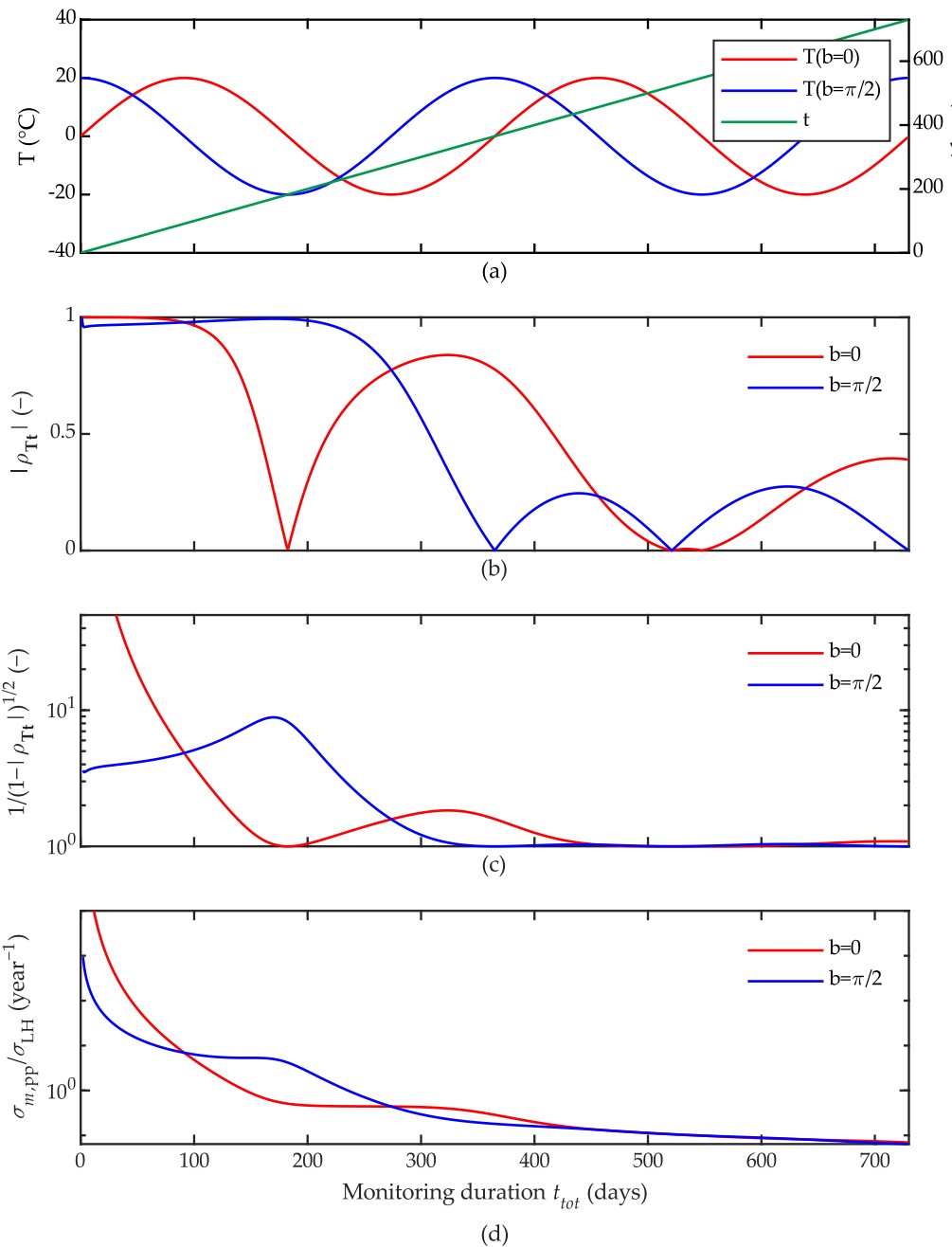

**Figure 1.** Impact of the time–temperature correlation on the pre-posterior uncertainty as the monitoring duration $t_{tot}$ increases from 1 to 730 days in the limit cases $b = 0$ (red lines) and $b = \pi/2$ (blue lines). (**a**) Time **t** with $N = 730$ and $f_s = 1/\text{day}$, and temperature **T** with $a = 20\,^\circ\text{C}$ and $c = 0\,^\circ\text{C}$; (**b**) the absolute value of the linear correlation coefficient $\rho_{\mathbf{Tt}}$ between **t** and **T**; (**c**) term $1/\sqrt{\left(1 - \rho_{\mathbf{tT}}^2\right)}$ of Equation (14); (**d**) the ratio between the pre-posterior uncertainty $\sigma_{m,\text{pp}}$ and the likelihood uncertainty $\sigma_{\text{LH}}$.

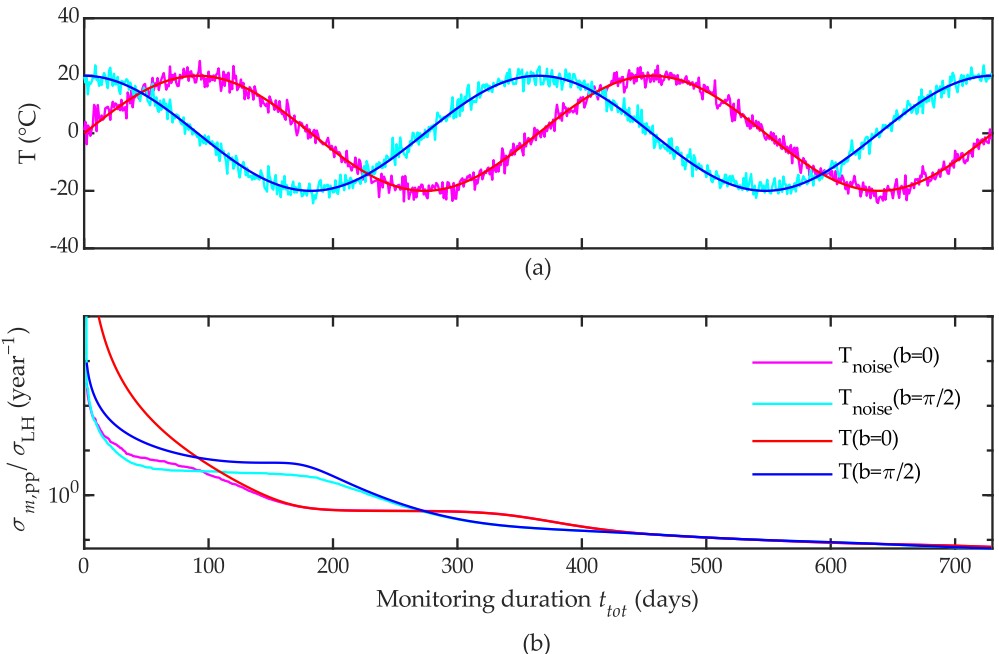

**Figure 2.** Impact of the day-by-day variation in temperature simulated as Gaussian noise $n_i(\sigma_{noise})$ on the pre-posterior uncertainty as the monitoring duration $t_{tot}$ increases from 1 to 730 days in the limit cases $b = 0$ and $b = \pi/2$. (**a**) Comparison between $\mathbf{T_{noise}}$ and $\mathbf{T}$; (**b**) comparison between $\sigma_{m,pp}(\mathbf{T},t)/\sigma_{LH}$ and $\sigma_{m,pp}(\mathbf{T_{noise}},t)/\sigma_{LH}$. Temperatures with $a = 20\,°C$, $c = 0\,°C$, and $\sigma_{noise} = 2\,°C$.

Figure 2b shows that the pre-posterior uncertainty calculated with the simulated experimental temperature function $\mathbf{T_{noise}}$ is smaller than the pre-posterior uncertainty calculated with the theoretical temperature function $T$. Indeed, the random variability $n_i(\sigma_{noise})$ in $\mathbf{T_{noise}}$ reduces the linear correlation between the temperature and time. As a result, the evaluation of $\sigma_{m,pp}/\sigma_{LH}$ based on the theoretical temperature $\mathbf{T}$ provides safer results compared to the scenario where it is based on some real temperature data affected by day-by-day variability.

*3.4. Impact of the Sampling Frequency*

Term $1/|\Delta\mathbf{t}|$ in Equation (14) is the reciprocal of the absolute value of vector $\Delta\mathbf{t}$, i.e., the reciprocal of the standard deviation of vector $\mathbf{t}$. We can express $1/|\Delta\mathbf{t}|$ as

$$\frac{1}{|\Delta\mathbf{t}|} = \frac{1}{\sqrt{\sum_{i=1}^{N}\left(\frac{1}{f_s}(i-1) - \frac{t_{tot}}{2}\right)^2}} = \frac{1}{\frac{1}{f_s}\sqrt{\frac{N}{12}(N^2-1)}} \qquad (26)$$

The term $1/|\Delta\mathbf{t}|$ depends on both the number of samples $N$ and the sampling frequency $f_s$, and behaves as the inverse of an inertial factor: as the monitoring duration and sampling frequency increase, $1/|\Delta\mathbf{t}|$ and $\sigma_{m,pp}$ decrease. Figure 3 shows the impact of the monitoring duration and sampling frequency on $1/\sqrt{(1-\rho_{\mathbf{tT}}^2)}$, $1/|\Delta\mathbf{t}|$ and $\sigma_{m,pp}$; it clearly shows that $1/\sqrt{(1-\rho_{\mathbf{tT}}^2)}$ does not depend on $f_s$, unlike $1/|\Delta\mathbf{t}|$.

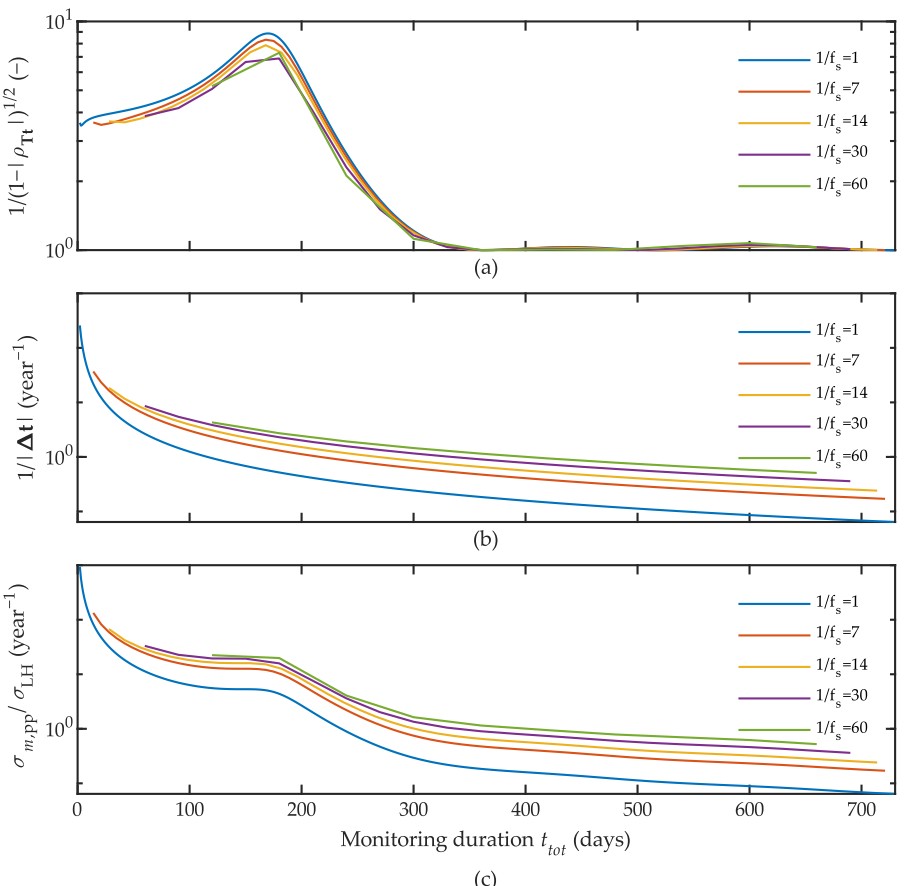

**Figure 3.** Impact of the sampling frequency $f_s$ on pre-posterior uncertainty $\sigma_{m,\mathrm{pp}}$. The graphs are plotted against the monitoring duration $t_{tot}$ from 1 to 730 days, for different time intervals $1/f_s$ between measurements, from 1 to 60 days. (**a**) Impact on term $1/\sqrt{(1-\rho_{\mathbf{tT}}^2)}$; (**b**) impact on term $1/|\Delta \mathbf{t}|$, where $b$ is set equal to $\pi/2$; (**c**) impact on $\sigma_{m,\mathrm{pp}}/\sigma_{\mathrm{LH}}$.

### 3.5. Impact of the Residual's Noise $\sigma_{LH}$

The residual's noise $\sigma_{\mathrm{LH}}$ works as an amplification coefficient on $\sigma_{m,\mathrm{pp}}$: as it decreases, $\sigma_{m,\mathrm{pp}}$ decreases. In order to reduce $\sigma_{\mathrm{LH}}$, and thus to obtain a more accurate estimation of $m$, we can choose high-performance sensors; in particular, the sensors' accuracy (i.e., the small random noise $\sigma_\varepsilon$ and $\sigma_T$) is critical when the model uncertainty $\sigma_{model}$ is small. However, under the assumption of statistically uncorrelated errors, the sensor accuracy can be less critical: high model uncertainties $\sigma_{model}$ strongly reduce the influence of sensor noise $\sigma_y$ and $\sigma_T$ on $\sigma_{\mathrm{LH}}$; therefore, using very accurate and expensive instrumentation does not drastically improve the monitoring effectiveness in the estimation of the trend parameter.

### 3.6. Impact of Prior Distributions

Finally, we can analyse the difference between Equations (12) and (14). Figure 4 shows a comparison between $\sigma_{m,\mathrm{pp}}$ with Gaussian prior parameter distributions (Equation (12)), and $\sigma_{m,\mathrm{pp}}$ neglecting prior parameter distributions (Equation (14)).

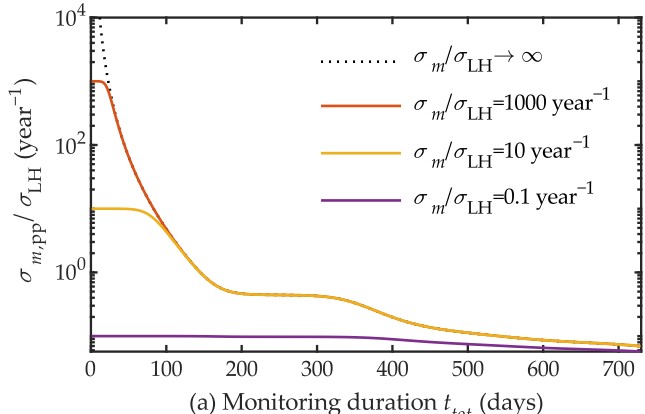 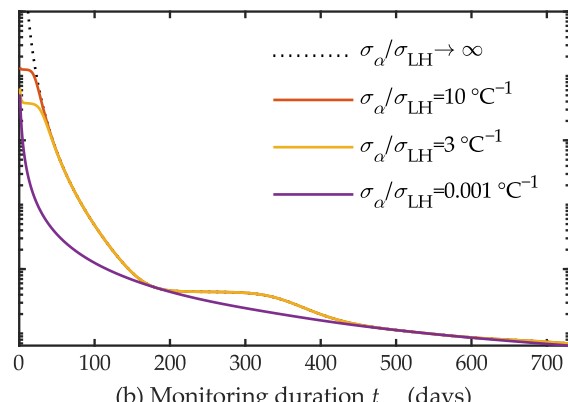

(a) Monitoring duration $t_{tot}$ (days)    (b) Monitoring duration $t_{tot}$ (days)

**Figure 4.** Impact of the prior distributions of parameters $\alpha$ and $m$ on the pre-posterior uncertainty. The graphs are plotted against the monitoring duration $t_{tot}$ from 1 to 730 days, $b = 0$, $f_s = 1$. (**a**) $\sigma_m/\sigma_{LH} \to \infty$, $\sigma_\alpha/\sigma_{LH}$ varies from 10 to 0.001 °C$^{-1}$; (**b**) $\sigma_\alpha/\sigma_{LH} \to \infty$, $\sigma_m/\sigma_{LH}$ varies from 1000 to 0.1 year$^{-1}$.

In Figure 4a, Equation (12) is plotted assuming different possible prior Gaussian distributions of parameter $m$ while the prior distribution of the thermal coefficient $\alpha$ is neglected ($\sigma_\alpha/\sigma_{LH} \to \infty$). In Figure 4b, Equation (12) is plotted assuming different possible prior Gaussian distributions of $\alpha$ while the prior distribution of $m$ is neglected ($\sigma_m/\sigma_{LH} \to \infty$). The standard deviations of the prior distributions range from highly uncertain to very accurate values: from $\sigma_m/\sigma_{LH} = 1000$ year$^{-1}$ to $\sigma_m/\sigma_{LH} = 0.1$ year$^{-1}$ in Figure 4a, and from $\sigma_\alpha/\sigma_{LH} = 10$ °C$^{-1}$ to $\sigma_\alpha/\sigma_{LH} = 0.001$ °C$^{-1}$ in Figure 4b.

When the prior distribution of the model parameters is highly uncertain (orange solid lines), Equation (12) provides the same results as Equation (14) (black dashed lines), apart from very short monitoring durations.

In contrast, when the prior distribution of $m$ is very accurate (the purple solid line in Figure 4a), $\sigma_{m,pp}$ stays almost constant and slightly decreases as the monitoring duration increases. Therefore, if the trend parameter $m$ is accurately known a priori, structural health monitoring is not necessary. When $\sigma_m/\sigma_{LH} \to 0$, Equation (12) becomes

$$\sigma_{m,pp}(\mathbf{T,t}) = 0 \; \forall \; N \tag{27}$$

On the other hand, when the prior distribution of $\alpha$ is very accurate (the purple solid line in Figure 4b), $\sigma_{m,pp}$ is completely independent from the linear correlation between the time and temperature. Indeed, if the thermal coefficient $\alpha$ is accurately known a priori, we can compensate the temperature effects deterministically. When $\sigma_\alpha/\sigma_{LH} \to 0$, Equation (12) becomes

$$\sigma_{m,pp}(\mathbf{T,t}) = \sigma_{LH} \frac{1}{\sqrt{\frac{\sigma_{LH}^2}{\sigma_m^2} + |\mathbf{\Delta t}|^2}} \tag{28}$$

Generally, for typical values of the prior uncertainty of the parameters, the prior Gaussian distributions affect the results of $\sigma_{m,pp}$ only for short monitoring periods. Consequently, as the monitoring duration increases, the prior distributions become less influential, and Equation (12) resembles Equation (14).

## 4. Colle Isarco Viaduct Case Study

The Colle Isarco Viaduct [32] is an Italian prestressed concrete highway bridge. It was erected in 1968, and opened to traffic in 1971. The viaduct consists of two structurally independent decks, both with 13 spans, for a total length of 1028 m. The main span is 163 m long, and is made of two symmetric reinforced concrete Niagara box girders, which support a 45 m-long suspended beam. At the end of each box girder is a 59 m-long cantilever, counterbalanced by a back arm with a length of 91 m. Each box girder is

composed of 33 cast-in-place segments with a depth varying from 10.93 m at the pier to 2.57 m at the edge. The thickness of the top slab is constant, at 0.29 m, while the bottom slab varies from 0.99 m to 0.12 m. A concrete with nominal class $R_{ck}$ = 450 kg/cm$^2$ was used for all of the cast-in-place elements of thepiers and girders. The initial prestressing was applied through 32 mm diameter Dywidag ST 85/105 threaded bars, with an initial jacking tension of 720 MPa. For each 59m-long cantilever, the longitudinal force above the pier was about 120 MN, and was provided by a total of 266 cables. Figure 5a shows a picture of the viaduct from pier 7 to 9. Figure 5b shows a longitudinal section of the viaduct between piers 7 and 10, as well as two cross-sections of the box girders.

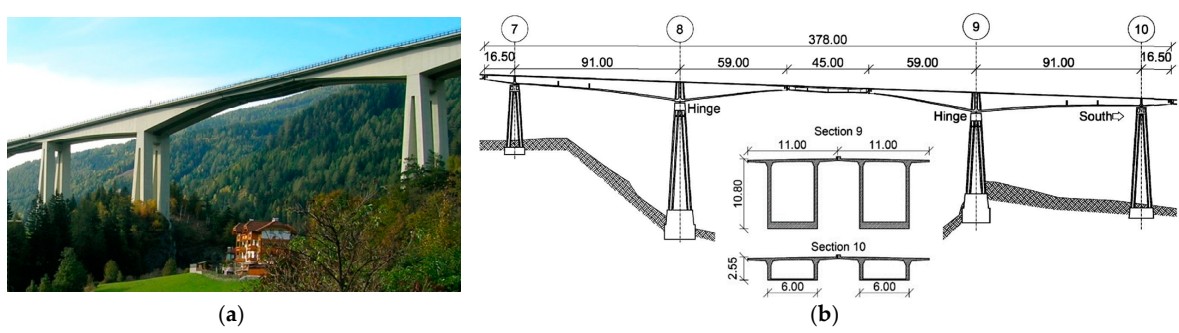

(**a**)  (**b**)

**Figure 5.** (**a**) The main span and northernmost cantilever of Colle Isarco Viaduct; (**b**) a longitudinal section of the viaduct between piers 7 and 10, and cross-sections of the box girders, with dimensions in meters.

The Colle Isarco Viaduct has been subjected to excessive creep and shrinkage phenomena since its construction. It has experienced a progressive abnormal deformation of the top and bottom slab, resulting in an abnormal deflection of the cantilever beams, which are mostly evident at the edge of the 59 m-long cantilever even several decades after its construction. This high sensitivity to creep and shrinkage phenomena is experienced by many other bridges of the same type, and has been investigated by several authors [23,35,36]. They suggest that the cause of such behaviour is the combination of creep phenomena, prestress tension losses, the huge difference between the top and the bottom slab in terms of thickness, the variation of the load condition during the construction phase, and maintenance works. Just a few years after its construction, the main span showed an excessive deflection trend, which resulted in a deflection of 100 mm in 1976 and 200 mm in 1984, while the design prediction was less than 20 mm. In 2014, through a retrofit work, the four box girders were equipped with an external post-tensioning system, which provides additional prestress through 212 0.6" diameter compact strands, with a jacking load of 213 kN. The additional longitudinal force produced above the piers was about 45 MN, which is almost 40% of the original prestress. The thickness of the top slab of the box girder was increased from 260 mm to 290 mm, to compensate for the additional post-tensioning force. This retrofit work reduced the deflection by 80 mm and changed the deflection drift from negative to positive. Details of the retrofit work can be found in the relevant design documentation [37].

The Colle Isarco Viaduct is currently monitored by three different technologies [22]. First, a topographic network measures the 3D displacements of the decks between pier 7 and pier 10. It consists of two stations, Leica Nova TM50, which collimate GPR112 prisms: 60 measurement points and 12 benchmarks. The topographic system was installed in 2014 before the retrofit works. Second, a network of 82 resistance temperature detectors (RTDs), TH-PT100, provided by Nova Metrix measures the local concrete temperature of the top and bottom slabs of the four cantilevers. The RTD network was installed in 2016. Third, 56 long-gauge fibre optic sensors (FOSs) measure the concrete local strain in the middle of the top and bottom slab. They are 12.1010 MuST deformation sensors, provided by Smartec SA. The FOSs network was installed in 2016 to monitor the long-

term effects of the 2014 post-tensioning intervention. In the present study, the focus is on strain and temperature measurements: Figure 6 shows the positions of the RTDs and FOSs on the northernmost cantilever of the northbound carriageway. The strain and temperature sensors are placed next to each other, and take one measurement every 15 min simultaneously. The local strain measured in the concrete slabs allows the effective calculation of the curvature of the girders, and the investigation of the causes of possible excessive long-term deformation trends. Conversely, the curvature calculated from the displacements provided by the topographic systems would be affected by severe errors due to the propagation of uncertainty. The local temperature of the concrete allows the compensation of the total strain measurements, which are severely affected by the response of the structure to daily and seasonal temperature variations.

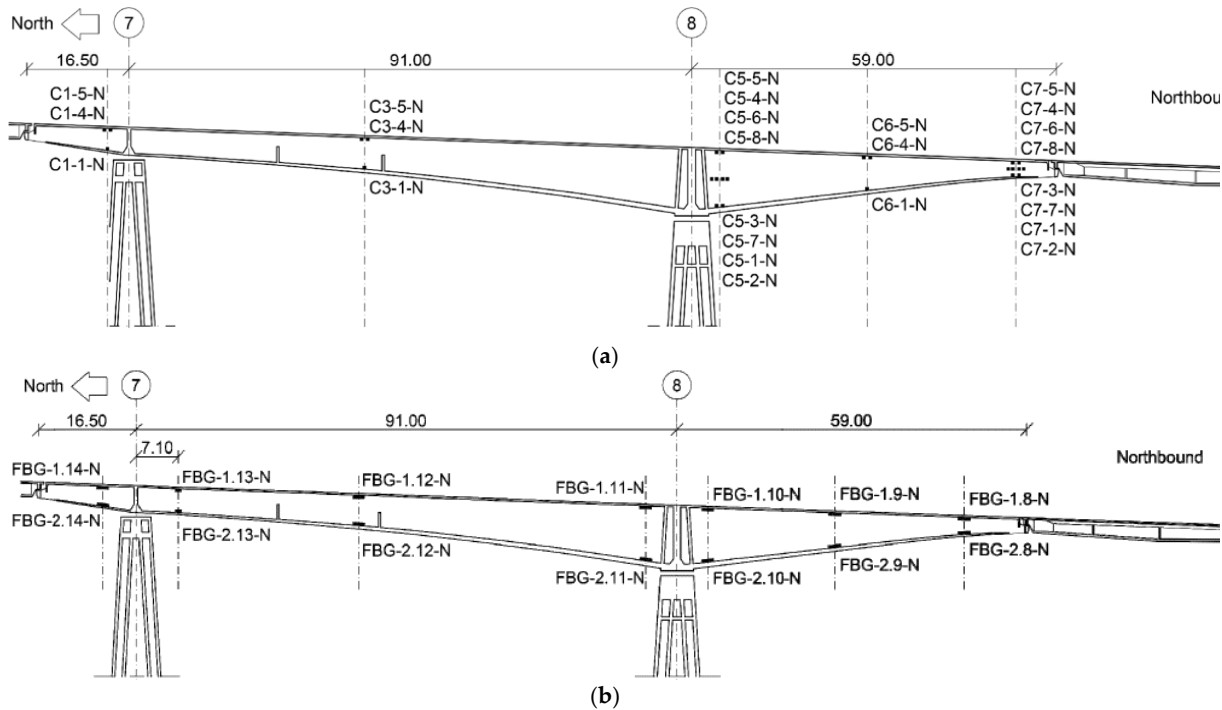

**Figure 6.** Positions of (**a**) the RTDs and (**b**) the FOSs on the northernmost cantilever of the northbound carriageway. The dimensions are given in meters.

## 5. Application to the Case Study

In this Section, we apply our approach to the Colle Isarco Viaduct case study; our aim is the evaluation of the expected uncertainty of the long-term strain trend of concrete, as compensated from temperature effects. In particular, we consider the strain and temperature monitoring system currently installed on the structure. First, we play the role of monitoring system designers, who chose the strain and temperature sensors and their location on the concrete box girders in 2016; our goal is the pre-posterior estimation of the expected uncertainty of the strain trend through Equation (12). Then, we play the role of monitoring data analysts, and we analyse the local strain and temperature recorded by the monitoring system; here, the goal is to infer a posteriori the posterior uncertainty based on the monitoring data acquired from the structure from April 2017 to February 2019. Finally, we validate the proposed approach by comparing the expected uncertainty with the posterior uncertainty, and we discuss the impact of the monitoring duration, monitoring start date, and temperature simulation.

### 5.1. Pre-Posterior Analysis: Expected Uncertainty in the Design Phase

Let us play the role of the monitoring system designer. It is the year 2016, and we are designing a local-strain monitoring system for the concrete box girders of the Colle Isarco



Viaduct. It must provide the local-strain linear trend $m$, as compensated from the seasonal temperature changes. We have already chosen the tentative strain and temperature sensors: their random noise is a zero-mean Gaussian variable with standard deviations $\sigma_\varepsilon$ and $\sigma_T$, respectively. In addition, we have already defined the sensor placement, see Figure 6, and the linear interpretation model, see Equation (1). Our goal is the pre-posterior estimation of the expected uncertainty $\sigma_{m,\mathrm{pp}}$ of the local strain linear trend $m$, in order to verify whether it is lower than a target value set by the viaduct manager, who oversees the infrastructure operation. We assume that once the monitoring system is up and running, it will acquire one local strain $\varepsilon_i$ and one temperature measurement $T_i$ per day, for $N$ days.

### 5.1.1. Simulation of the Expected Time and Temperature

First, we must define the elements within the time $t_i$ and temperature $T_i$ vectors.

We simulate time through Equation (16), and we set the sampling frequency as one measurement per day, $f_s = 1$ day$^{-1}$. We obtain the same vector $\mathbf{t}$ in Figure 1a. We wish to study how $\sigma_{m,\mathrm{pp}}$ changes with the monitoring duration $t_{tot}$ varying from 3 to 600 days.

Then, we simulate the expected seasonal temperature variation in two different ways: a sine function $\mathbf{T}$, defined as Equation (17), and a sine function with Gaussian noise $\mathbf{T_{noise}}$, defined as Equation (25). In order to represent two different start dates of the monitoring, we set two cases of the phase-parameter: $b = \pi/2$ and $b = 0$. In the first case, the monitoring period starts when the seasonal temperature reaches approximately its maximum stationarity point (during the summer season); in the second case, it starts in proximity to its point of inflection (during the spring season).

### 5.1.2. Estimation of the Measurements and Model Uncertainties

We must calculate the likelihood-function uncertainty $\sigma_{\mathrm{LH}}$ through Equation (22), where the measurement uncertainty $\sigma_y$ is represented by the standard deviation $\sigma_\varepsilon$ of the strain-sensors' zero-mean Gaussian random noise:

$$\sigma_{\mathrm{LH}} = \sqrt{\alpha^2 \cdot \sigma_T^2 + \sigma_\varepsilon^2 + \sigma_{model}^2} = 21 \ \mu\varepsilon \tag{29}$$

where:

- $\alpha = 10 \ \mu\varepsilon/{}^\circ\mathrm{C}$ is the local-concrete thermal-expansion coefficient at 20 °C [21].
- $\sigma_\varepsilon = 2 \ \mu\varepsilon$ is the accuracy of the strain sensors.
- $\sigma_T = 0.5 \ {}^\circ\mathrm{C}$ is the accuracy of the temperature sensors.
- $\sigma_{model} = 20 \ \mu\varepsilon$, based on similar case studies [7,22].

### 5.1.3. Expected Uncertainty

We perform the pre-posterior estimation of the expected uncertainty $\sigma_{m,\mathrm{pp}}$ of the local strain linear trend $m$ through Equation (12). We use the likelihood-function uncertainty $\sigma_{\mathrm{LH}}$ defined in Equation (29) and the time and temperature vectors defined in Section 5.1.1. In particular, we calculate $\sigma_{m,\mathrm{pp}}$ with the two simulated temperature vectors, $\mathbf{T}$ and $\mathbf{T_{noise}}$, and we study how $\sigma_{m,\mathrm{pp}}$ changes as the monitoring duration $N$ increases from 3 to 600 days. Figure 7 shows the results, which we will discuss in Section 5.3, along with the results on the real uncertainty $\sigma_{m \mid \varepsilon}$ inferred a posteriori based on the monitoring observations.

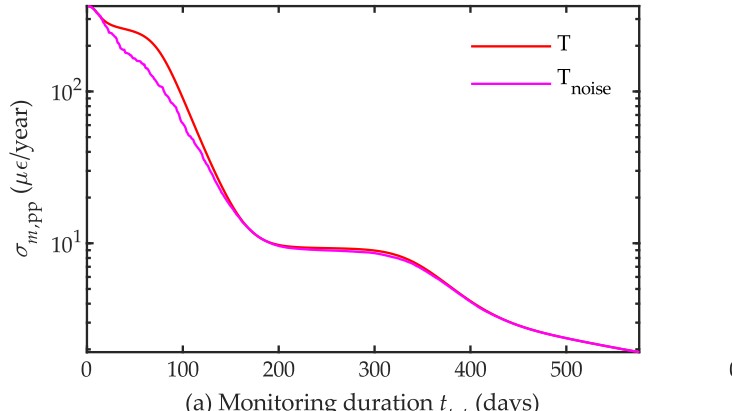
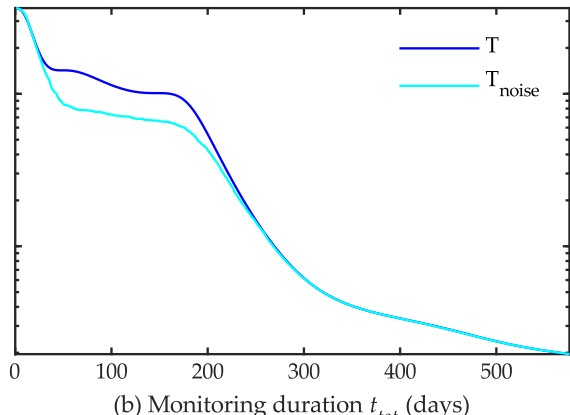

**Figure 7.** Expected uncertainty $\sigma_{m,\text{pp}}$ of the linear-trend parameter $m$ estimated in the design phase. Phase parameter b is set as (**a**) b = 0 and (**b**) b = $\pi/2$. The monitoring duration $t_{tot}$ varies from 3 to 600 days.

### 5.2. Posterior Analysis: Real Uncertainty Based on the Monitoring Data

Let us play the role of the monitoring data analyst. The structural health monitoring system has been running for 4 years. We analyse the measurements of local-strain $\varepsilon$ and the temperature acquired from April 2017 to February 2019 from the bottom slab of the northernmost box girder C5-N by strain sensor FBG-2.10-N and temperature sensors C5-1-N and C5-2-N (see Figure 6). We analyse the mean value of the temperature data recorded by the two sensors, which we label $\mathbf{T}_{\text{str}}$. Our goal is now to infer a posteriori the posterior uncertainty $\sigma_{m\,|\,\varepsilon}$ of the local strain linear trend $m$, based on these monitoring data, and to study how it changes when the monitoring duration $N$ increases from 3 to 600 days.

Under the hypotheses Hp. 1 and Hp. 2 (linear-Gaussian model), we define the prior Gaussian probability distribution of model parameters $m$ and $\alpha$: parameter $m$ has prior mean value $\mu_m = 0$ and prior standard deviation $\sigma_m = 365\ \mu\varepsilon/\text{year}$; parameter $\alpha$ has prior mean value $\mu_\alpha = 10$ and prior standard deviation $\sigma_\alpha = 3\ \mu\varepsilon/^\circ\text{C}$. Moreover, we consider $\sigma_{\text{LH}}$ as an additional model parameter to be estimated a posteriori, and we assume its prior probability distribution to be uniform. Then, we perform a Bayesian parameter estimation through a Markov chain Monte Carlo method based on the Metropolis Hasting algorithm [38]. Through an iterative process, this method estimates the posterior probability distribution of parameters p($m\,|\,\varepsilon$) and p($\alpha\,|\,\varepsilon$), as well as the distribution of the residual p($z\,|\,\varepsilon$), for any chosen monitoring duration. Finally, we perform temperature compensation on the total strain measurements by removing the thermal-strain component of the model, $\alpha\cdot\mathbf{T}_{\text{str}}$, and isolating the strain trend due to long-term effects $m\cdot\mathbf{t}$. Figure 8a,b show the strain and temperature measurements recorded from 20 July 2017 to 17 February 2019, respectively; Figure 8c shows the temperature-compensated strain-data $\varepsilon - \alpha\cdot\mathbf{T}_{\text{str}}$, as well as the linear strain trend $m\cdot\mathbf{t}$.

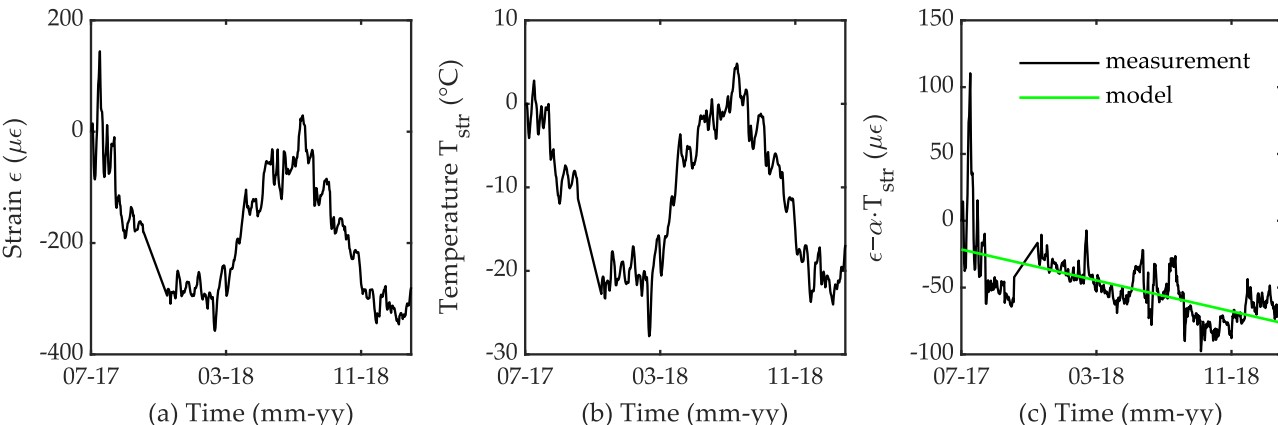

**Figure 8.** (**a**) 600 days of strain measurements from sensor FBG-2.10-N; (**b**) mean values of 600 days of temperature measurements from sensors C5-1-N and C5-2-N; (**c**) temperature-compensated strain measurements and the posterior local-strain linear-trend $m \cdot t$ over 600 days. Measurements of each sensor are plotted subtracting the first value.

We are interested in studying the ways in which the model parameter $m$, its uncertainty $\sigma_{m\,|\,\varepsilon}$, and the uncertainty of the likelihood function $\sigma_{\text{LH}\,|\,\varepsilon}$ a posteriori change as the monitoring duration increases from 3 to 600 days. In order to obtain the same initial conditions as in the design phase (Section 5.1.1) in terms of the monitoring start date, we fit $\mathbf{T}_{\text{str}}$ with a sinusoidal function defined as Equation (17), and we identify the timestamps corresponding to the maximum stationarity point and the inflection point: the first corresponds approximately to 20 July, and the second corresponds to 20 April. Therefore, we perform parameter estimation and temperature compensation with two sets of monitoring data: the first starts on 20 April 2017, corresponding to setting b = 0 in the design simulated temperature function, while the second starts on 20 July 2017, corresponding to setting b = π/2 in the design simulated temperature function. We repeat the analysis multiple times, changing monitoring duration $t_{tot}$ from 3 to 600 days. Figure 9a,b show the results in terms of $m$ and $\sigma_{\text{LH}\,|\,\varepsilon}$, with monitoring data starting on 20 July 2017. They are plotted along with their 0.01 and 0.99 percentile to highlight the interval of confidence.

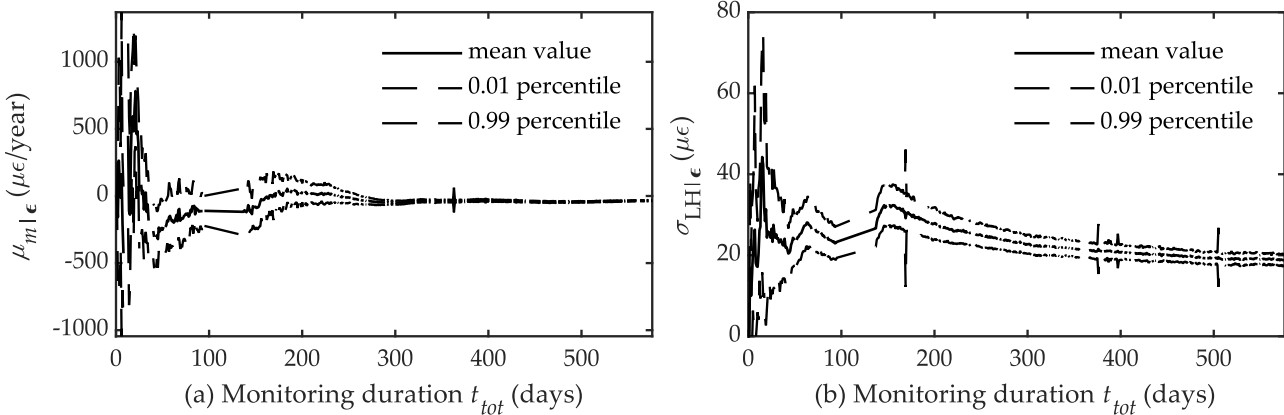

**Figure 9.** (**a**) Mean value and percentiles 0.01 and 0.99 of parameter $m$ a posteriori; (**b**) mean value and percentiles 0.01 and 0.99 of the likelihood-function uncertainty $\sigma_{\text{LH}\,|\,\varepsilon}$ a posteriori.

Figure 9a shows the posterior mean value of $m$, $\mu_{m\,|\,\varepsilon}$, as the monitoring duration increases. It varies as the monitoring duration increases during the first year of monitoring; for longer monitoring durations, it settles around a constant value. The reason is that $m$ is affected by a higher uncertainty due to the time–temperature correlation during the first

period of monitoring, while this correlation decreases and becomes uninfluential for longer monitoring durations, as we observed in Section 3.3.

Figure 9b shows the posterior mean value of the likelihood-function uncertainty $\mu_{\sigma LH \mid \varepsilon}$ as the monitoring duration increases. Like the trend parameter $m$, $\sigma_{\text{LH}}$ stabilizes at an approximately constant value after around one year of monitoring; this indicates that the linear interpretation model defined in Equation (1) is appropriate to approximate the long-term structural response.

Figure 10 shows the posterior uncertainty $\sigma_{m \mid \varepsilon}$ as the monitoring duration increases for the two limit cases of the monitoring start date.

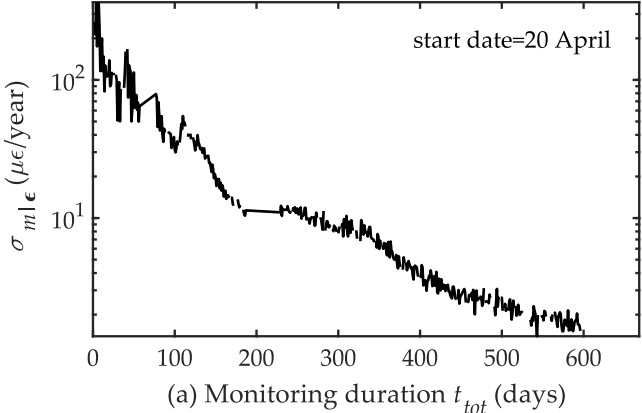
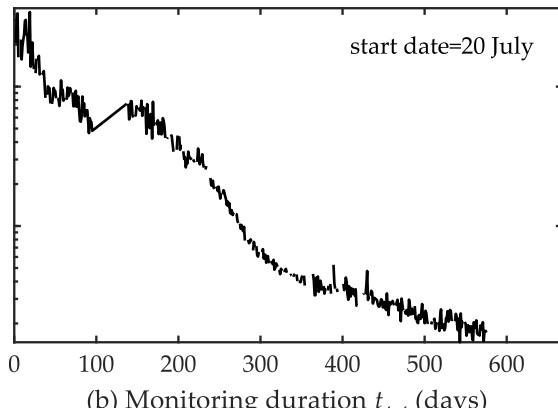

(a) Monitoring duration $t_{tot}$ (days)

(b) Monitoring duration $t_{tot}$ (days)

**Figure 10.** Real uncertainty $\sigma_{m \mid \varepsilon}$ of the linear-trend parameter $m$, estimated a posteriori based on the monitoring data. The monitoring start dates are (**a**) 20 April and (**b**) 20 July.

*5.3. Discussion of the Results*

Here, we validate the proposed approach by comparing the expected uncertainty with the posterior uncertainty, and discuss the impact of the monitoring duration, the monitoring start date, and the temperature simulation. Finally, we compare our approach.

5.3.1. Expected vs. Real Uncertainty

Based on the case study results, we aim to validate the proposed approach for the estimation of the pre-posterior uncertainty of the structural long-term linear response trend. Therefore, we compare the expected uncertainty $\sigma_{m,\text{pp}}$ with the posterior uncertainty $\sigma_{m \mid \varepsilon}$, and we discuss their difference as we change the monitoring duration, monitoring start date, and temperature simulation.

Figure 11 shows the comparison between the pre-posterior linear trend uncertainty $\sigma_{m,\text{pp}}$ estimated with **T** and **T$_{\text{noise}}$**, and the linear trend uncertainty calculated a posteriori, $\sigma_{m \mid \varepsilon}$. They are plotted against monitoring duration $t_{tot}$; we reported both the limit scenarios of monitoring start date: 20 April, corresponding to $b = 0$ in the design simulated temperature function, and 20 July, corresponding to $b = \pi/2$ in the design simulated temperature function. Here, $\sigma_m$ is 1 $\mu\varepsilon$/day = 365 $\mu\varepsilon$/year, which is the value used in the Bayesian parameter estimation described in Section 5.2.

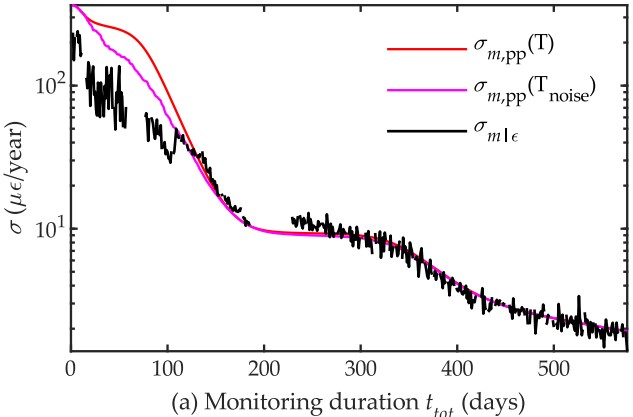 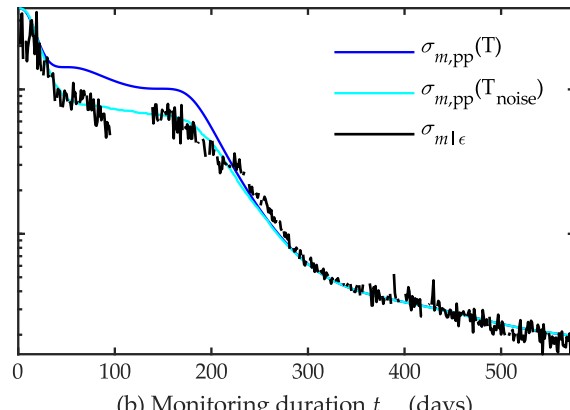

(a) Monitoring duration $t_{tot}$ (days)          (b) Monitoring duration $t_{tot}$ (days)

**Figure 11.** Pre-posterior $\sigma_{m,\mathrm{pp}}$ and posterior $\sigma_{m\,|\,\varepsilon}$ linear trend uncertainty plotted against the monitoring duration $t_{tot}$. The monitoring start dates are (**a**) 20 April, $b = 0$, and (**b**) 20 July, $b = \pi/2$. The monitoring duration $t_{tot}$ varies from 3 to 600 days.

We note that the expected—pre-posterior—uncertainty estimated with **T** tends to overestimate the real—posterior—uncertainty for short monitoring durations, while they become qualitatively similar for longer monitoring durations. In contrast, the expected uncertainties estimated with both **T$_{\mathbf{noises}}$** is closer to the real uncertainty even for short monitoring durations. Such results confirm what we observed in Section 3.3: the expected uncertainty is smaller when calculated with **T$_{\mathbf{noise}}$** for short monitoring durations, because the random variability $n_i(\sigma_{noise})$ in **T$_{\mathbf{noise}}$** reduces the correlation between the temperature and time. As a result, within the calculation of the expected uncertainty $\sigma_{m,\mathrm{pp}}$ with **T$_{\mathbf{noise}}$**, the posterior uncertainty of the trend-parameter $\sigma_{m\,|\,\varepsilon}$ might be underestimated, depending on the chosen value of $\sigma_{noise}$. In contrast, the expected uncertainty $\sigma_{m,\mathrm{pp}}$ calculated with **T** always slightly overestimates the posterior one, so that its use is safe for the design purpose.

In order to quantify the difference between the expected and real uncertainty of a model parameter, and to validate the proposed approach, we use the concept of monitoring system effectiveness [3] in the estimation of the model parameter $m$. The monitoring effectiveness, $\eta$, expresses the extent to which the monitoring observations improve the knowledge of the structure behaviour; in other words, the extent to which the parameter uncertainty a posteriori is lower than a priori. We define the expected effectiveness of the monitoring system, $\eta_{\exp}$, as the ratio between the prior uncertainty $\sigma_m$ and the pre-posterior uncertainty $\sigma_{m,\mathrm{pp}}$. We define the real effectiveness of the monitoring system, $\eta_{\mathrm{real}}$, as the ratio between the prior uncertainty $\sigma_m$ and the posterior uncertainty $\sigma_{m\,|\,\varepsilon}$.

$$\eta_{\exp} = \sigma_m / \sigma_{m,\mathrm{pp}} \tag{30}$$

$$\eta_{\mathrm{real}} = \sigma_m / \sigma_{m\,|\,\varepsilon} \tag{31}$$

The inverse of the monitoring effectiveness, $1/\eta$, expresses the ineffectiveness of the monitoring system: the closer $1/\eta$ is to zero, the higher the reduction in the parameter uncertainty thanks to the monitoring data; the closer $1/\eta$ is to 1, the lower the reduction in the parameter uncertainty thanks to the monitoring data. Figure 12 shows the expected and real ineffectiveness of the monitoring system, $1/\eta_{\exp}$ and $1/\eta_{\mathrm{real}}$, respectively.

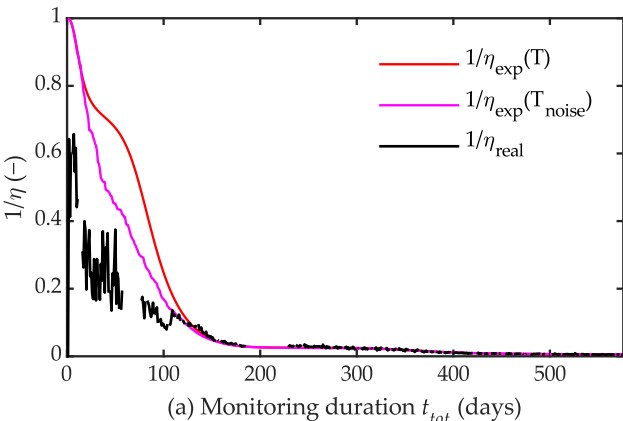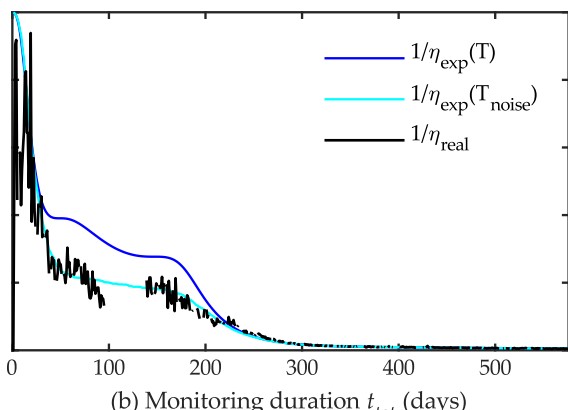

**Figure 12.** Expected $1/\eta_{\text{exp}}$ and real $1/\eta_{\text{real}}$ ineffectiveness of the monitoring system. The $1/\eta_{\text{exp}}$ is calculated with **T** and **T$_{\text{noise}}$**. The monitoring start dates are (**a**) 20 April, b = 0, and (**b**) 20 July, b = $\pi/2$. The monitoring duration $t_{tot}$ varies from 3 to 600 days.

As the monitoring duration increases, both the expected and the real monitoring ineffectiveness $1/\eta$ decrease. The monitoring start date influences the monitoring duration required before $1/\eta$ stabilizes around zero. In particular, a monitoring system that starts measuring in summer reaches $1/\eta \sim 0$ after around 250 days; in contrast, a monitoring system that starts measuring in spring reaches $1/\eta \sim 0$ after around 150 days. A few days' variations depend on the simulated temperature used.

Finally, we quantify the error $e_{m,\text{pp}}$ in the pre-posterior estimation of the linear trend uncertainty $\sigma_{m,\text{pp}}$ as the difference between the expected and real ineffectiveness of the monitoring system.

$$e_{m,\text{pp}} = 1/\eta_{\text{exp}} - 1/\eta_{\text{real}} \tag{32}$$

Figure 13 shows how the error $e_{m,\text{pp}}$ changes as the monitoring duration increases.

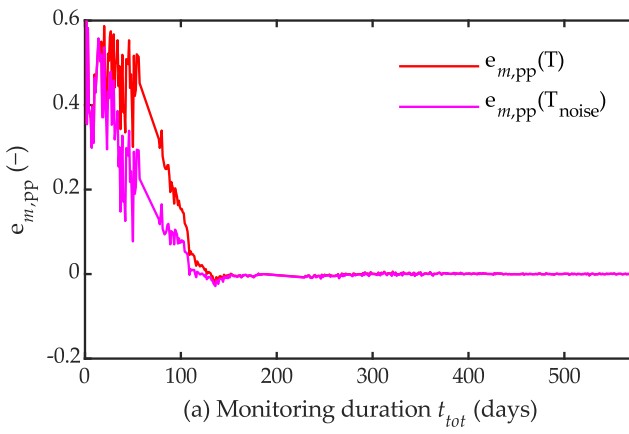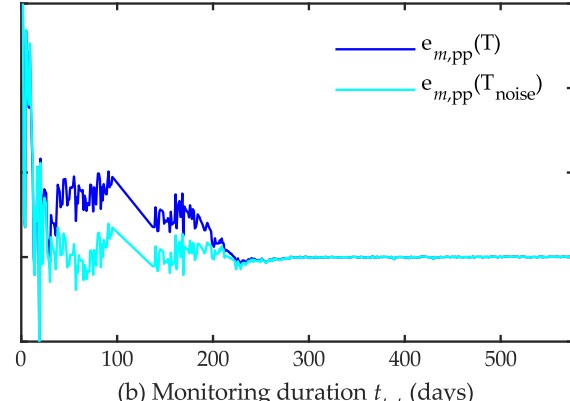

**Figure 13.** Error $e_{m,\text{pp}}$ between the expected and the real uncertainty of the parameter *m*. The expected uncertainty is calculated with **T** and **T$_{\text{noise}}$**. The monitoring start dates are (**a**) 20 April and (**b**) 20 July. $\sigma_m = 1\ \mu\varepsilon/\text{day} = 365\ \mu\varepsilon/\text{year}$.

The error $e_{m,\text{pp}}$ may have negative values according to the simulated temperature function used in the estimation of $\sigma_{m,\text{pp}}$ [39]; for positive values of $e_{m,\text{pp}}$, the posterior uncertainty $\sigma_{m\,|\,\varepsilon}$ is overestimated, while for negative values of $e_{m,\text{pp}}$, the posterior uncertainty $\sigma_{m\,|\,\varepsilon}$ is underestimated. In our case, when the monitoring starts in the spring, $e_{m,\text{pp}}$ is mostly positive with any simulated temperature function; it progressively decreases and zeroes after around 150 days. On the other hand, when the monitoring starts during

summer, $e_{m,pp}$ has some negative values in the evaluation with $\mathbf{T_{noise}}$; the value of $e_{m,pp}$ remains constantly equal to zero after a longer monitoring period of around 250 days.

We note that the expected uncertainty estimated with $\mathbf{T}$ is less accurate than the one with $\mathbf{T_{noise}}$, but it is always positive; the expected uncertainty estimated with our approach and with $\mathbf{T}$ always overestimates the real uncertainty. In this case, we may look at $e_{m,pp}$ as a safety factor: the monitoring system will never provide the key parameter with posterior uncertainty higher than what is expected. In contrast, the expected uncertainty estimated with $\mathbf{T_{noise}}$ is more accurate, but can underestimate the real uncertainty; this confirms again that the day-by-day variation of $\mathbf{T_{noise}}$ reduces the time–temperature correlation.

5.3.2. Proposed Approach vs. Previous Studies

Our approach is in line with many previous studies in the literature regarding the use of a linear interpretation model to combine the mechanical response with the temperature effects. Such a model can be used to effectively perform the temperature compensation of monitoring data: the effectiveness of this choice has been validated for mechanical responses such as strain measurements [40], ultrasonic measurements [41], vibrational measurements [27,31], and cracks opening [31]. The linear temperature compensation of the monitoring data has proven to be necessary to detect whether damage is present or not based on the measured dynamic or static characteristics of a monitored system [13]. Moreover, our approach is similar to that which is currently used for the temperature compensation of the sensor response [39,42–44], rather than the structural response. Indeed, the response of sensors is also sensible to temperature variation; therefore, sensor measurements must compensate for temperature effects before being used for structural assessment.

As far as the design of a structural health monitoring system is concerned, there are many previous studies regarding the optimal sensor placement based on the maximisation of the information acquired or on the maximisation of the value of information acquired for structural management purposes. However, to our best knowledge there aren't any previous studies regarding the estimation of the expected uncertainty of structural response trends accounting for temperature compensation for monitoring system design purposes.

Concerning the design of structural health monitoring systems, there are many previous studies regarding the optimal sensor placement based on the maximisation of the information acquired or on the maximisation of the value of information acquired for structural management purposes. However, to our best knowledge there aren't any previous studies on the estimation of the expected uncertainty of structural response trends accounting for temperature compensation for design purposes.

The main advantage of our approach is its ease of use by the majority of civil engineers, who typically have a solid background in structural design but not necessarily in statistics and probability. They can apply our algorithm to predict the performance of a tentative monitoring system in the design phase, even without being familiar with Bayesian probability. This is an important step forward for the extension of structural health monitoring to a higher number of bridges. Indeed, the effectiveness of a tentative monitoring system can be practically evaluated by comparing the expected uncertainty (the monitoring system capacity) resulting from the proposed algorithm with a target value required by the infrastructure operator (monitoring system demand). This corresponds to the comparison between the structural capacity and structural demand in structural design.

On the other hand, the main disadvantage of the proposed algorithm is its tendency to overestimate the real uncertainty of the trend-parameter for short monitoring periods, which might require a stronger performance of the sensors than is really necessary. This issue can be mitigated by choosing the simulated temperature $\mathbf{T_{noise}}$, which models the observed variation in the real temperature between two consecutive days, rather than the sinusoidal temperature $\mathbf{T}$. However, the use of $\mathbf{T_{noise}}$ requires the statistical estimation of an appropriate value of $\sigma_{noise}$ based on the values of the local temperature. Finally, the uncertainty of the linear model $\sigma_{model}$ should also be statistically estimated. It depends on the trend parameter to be identified (e.g., the strain trend, displacement trend, frequency trend,

crack amplitude trend) and on the structure to be monitored (e.g., prestressed concrete bridge, cable-stayed bridge, arch bridge). Concerning prestressed concrete highway bridges and long-term concrete strain trends, a value of $\sigma_{model} = 20$ µε is generally acceptable.

## 6. Conclusions

When designing a monitoring system, we need to predict beforehand the uncertainty of the key performance parameters we expect to achieve after the monitoring is performed. In this paper, we derived a simple-to-use formulation that allows us to calculate the expected uncertainty of a long-term structural response trend based on monitoring data compensated from temperature effects. This formulation does not depend on the response measurements; therefore, it can be used by the designer to validate the performances of a tentative monitoring strategy before the measurements are actually available. The formulation only requires us to make reasonable assumptions on the sampling timestamp vector, the expected temperature variation and the residual noise. The timestamp vector depends on the sampling frequency chosen by the designers. The temperature can be roughly, but effectively, simulated with a sine function with a period of 1 year. The residual error can be predicted by combining sensor noise and the approximation error of the interpretation model. According to the formulation, the uncertainty of a linear drift depends essentially on these factors:

- sensor accuracy, usually provided in the technical datasheets of the sensors;
- the correctness of the interpretation model, i.e., how well the model fits the actual structural response;
- the monitoring duration, and, to a minor extent, the sampling frequency;
- the degree of correlation between the time and temperature records, i.e., the extent to which the temperature record confuses with a straight line.

The formulation predicts that, for monitoring periods of a few months, the uncertainty is particularly sensitive to the linear correlation between the time and temperature, as temperature can be approximated to a linear trend. In particular, the fitting quality changes significantly depending on whether the monitoring start date is a solstice or an equinox. Beyond the first year of monitoring, the confusion between the time and temperature is negligible, and the quality of the monitoring primarily depends on the monitoring duration.

We validated our approach by applying the formulation to a real-life case study, the Colle Isarco Viaduct. This is one of the longest prestressed concrete bridges in the European Alpine region, and it is currently equipped with an optical fibre sensor network and resistance temperature detectors. We focused on a cross-section of the box girder of the bridge, and we estimated the expected uncertainty of the long-term strain trend, purged of seasonal temperature effects, with our approach. Then, we verified whether the pre-posterior estimation of uncertainty is consistent with its posterior estimation, based on the actual monitoring data.

We observed that the expected uncertainty accurately predicts the posterior uncertainty after 150–250 days of monitoring data. For shorter monitoring durations, the prediction accuracy depends on the way in which the expected temperature record was been simulated. In particular, the pre-posterior uncertainty estimated assuming an ideal sinusoidal temperature record always slightly overestimates the actual posterior uncertainty; therefore, its use is appropriate for the design purpose. Adding Gaussian noise to the simulated sine temperature allows the prediction of the posterior uncertainty with a smaller error, but it may underestimate it; therefore, its use could be less appropriate for the design purpose.

The proposed approach is an important step forward for the extension of structural health monitoring to a higher number of bridges. Indeed, it can easily be used by civil engineers to quantify the expected performance of a monitoring solution in its design phase, and it does not require a solid background in statistics and probability. It allows the verification beforehand of the effectiveness of a monitoring solution with the consolidated logical approach for structural design: capacity > demand.

Its validation for other types of structural response trends (e.g., displacement, rotation, frequency variation, crack propagation) will be further investigated in future research studies. Moreover, future research will address the estimation of the expected uncertainty, accounting for temperature compensation with a number of temperature sensors.

**Author Contributions:** Conceptualization, D.T. and D.Z.; methodology, V.F.C. and D.T.; software, V.F.C.; validation, D.Z.; formal analysis, V.F.C. and D.T.; investigation, V.F.C. and F.P.; resources, D.Z.; data curation, V.F.C.; writing—original draft preparation, V.F.C. and D.T.; writing—review and editing, D.T.; visualization, V.F.C. and F.P.; supervision, D.T. and D.Z.; project administration, D.T.; funding acquisition, D.T. and D.Z. All authors have read and agreed to the published version of the manuscript.

**Funding:** This research was funded by FONDAZIONE CARITRO CASSA DI RISPARMIO DI TRENTO E ROVERETO, grant number 2021.0224, ReLUIS in the frame of "Accordo tecnico di attuazione dell'accordo ex Art. 15 Legge 7 agosto 1990, N. 241 between CSLLPP and ReLUIS", and Autostrada del Brennero SpA/Brennerautobahn AG.

**Data Availability Statement:** The data that support the findings of this study are available on request from the corresponding author [D.T.]. Restrictions apply to the availability of these data, which were used under license for this study.

**Acknowledgments:** The work presented in this paper was carried out under the research agreement between Autostrada del Brennero SpA/Brennerautobahn AG and the University of Trento. The financial contribution of Autostrada del Brennero is acknowledged, together with that of the Fondazione CARITRO Cassa di Risparmio Trento e Rovereto and ReLUIS.

**Conflicts of Interest:** The authors declare no conflict of interest. The funders had no role in the design of the study; in the collection, analyses, or interpretation of data; in the writing of the manuscript, or in the decision to publish the results.

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
