# Peer review of "Designing a Structural Health Monitoring System Accounting for Temperature Compensation"

_infrastructures, doi:10.3390/infrastructures7010005_

Round 1
Reviewer 1 Report
The authors propose a framework to evaluate the effectiveness of a monitoring system accounting for temperature compensation in the paper. It applies to the design process of a tentative structural health monitoring system for civil infrastructures.
The topic is very interesting. The manuscript structure is well. The manuscript also has some of its technical merits. It should fall into Journal Infrastructures. However, some minor questions have to pay attention to:
- There are several typos in the manuscript. Recommend the authors should check the manuscript before submission.
- Some signs are displayed in part, such as radical signs in 257, 265, 276. It needs to be adjusted to an appropriate size.
- In Figure 9, the curve lines of 0.01 percentile and 0.99 percentile aren’t displayed clearly. Recommend the authors should increase the line width to redraw the two lines.
- In line 565, the authors used σm|pp in the text. Then, the authors used σm, pp in Figure 11. Please check it.
- The authors regard subscript as the same height of the variable in some images. Please check it.
- The discussion section needs to be improved. The author has not pointed out the disadvantage of the proposed algorithm or a comparison of enough previous literature. Recommend the author should extend your discussion.
- The conclusions are more like a discussion. The authors need to improve the section and give some valuable conclusions for real engineering. In addition, the author might enrich your conclusions. It might also suggest future research.
Hopefully, this will help in the revision of the manuscript.
Author Response
We thank the Reviewer for their valuable comments and we list our response below.
- There are several typos in the manuscript. Recommend the authors should check the manuscript before submission.
We have carefully double-checked the manuscript and corrected all the typos.
- Some signs are displayed in part, such as radical signs in 257, 265, 276. It needs to be adjusted to an appropriate size.
We corrected the radical sign where suggested and elsewhere, including: lines 219, 262, 270, 281, 286, 288, 298, 299, 336, 337, 341, Equations 14, 29.
- In Figure 9, the curve lines of 0.01 percentile and 0.99 percentile aren’t displayed clearly. Recommend the authors should increase the line width to redraw the two lines.
We increased the width of the 0.01 and 0.99 percentile as recommended.
- In line 565, the authors used σm|pp in the text. Then, the authors used σm, pp in Figure 11. Please check it.
Thank you for noticing it, σm,pp is the right symbol. We changed it also elsewhere, including: 574, 581, 590, 592, and 625.
- The authors regard subscript as the same height of the variable in some images. Please check it.
We changed Figure 2, 3, 7, 8, 11, 12, 13 fixing the subscripts heigh.
- The discussion section needs to be improved. The author has not pointed out the disadvantage of the proposed algorithm or a comparison of enough previous literature. Recommend the author should extend your discussion.
We added a full section, Section 5.3.2, to discuss comparison between our approach and previous studies, pointing out pros and cons of our method. Points of discussion include:
- The proposed linear interpretation model is in line with a number of previous studies about temperature compensation of strain, ultrasonic, vibrational and crack-opening measurements.
- Our approach is similar to what is currently used for temperature compensation of sensors’ response is sensors design.
- The main advantage of our approach is its ease of use by most civil engineers, who typically have a solid background in structural design but not necessarily in statistics and probability.
- The main drawback of the proposed formulation is it tends to overestimate the actual uncertainty of the trend-parameter for short monitoring periods. Moreover, the uncertainty of the linear model σmodel should also be statistically estimated.
- The conclusions are more like a discussion. The authors need to improve the section and give some valuable conclusions for real engineering. In addition, the author might enrich your conclusions. It might also suggest future research.
We added a paragraph about how real-life engineers could benefit from this method to effectively design a monitoring system: verify beforehand the effectiveness of a monitoring solution with the consolidated logical approach for structural design: capacity > demand.
Also, we included hints about possible further research on this topic, such as:
- validation for other types of structural response trends (e.g., displacement, rotation, frequency variation, crack propagation).
- estimation of the expected uncertainty accounting for temperature compensation with multiple number of temperature sensors.
Reviewer 2 Report
This article needs some modifications to be suitable for this journal. I suggest major revision for this paper. The main comments are:
- The paper is technically interesting; however, the novelty of this paper should be further justified and to establish the contributions to the new body of knowledge (mainly in the abstract and introduction sections).
- Abstract section should be improved considering the following structure: introduction, problem statement, methodology, results, and conclusion.
- In the Introduction section, the authors should improve the research background, the review of significant works in the specific study area, the knowledge gap, the problem statement, and the novelty of the research.
- Introduction is not organized well. The reviewer cannot readily see the significance of this study compared to the previous works. The literature review should be extended to more recently published works available in the literature. Some of the previous studies that were recently reported but aren’t in your literature review are:
(a) Effect of column base strength on steel portal frames in fire, M Rahman, JBP Lim, Y Xu, R Hamilton, T Comlekci, D Pritchard, Proceedings of the Institution of Civil Engineers-Structures and Buildings ….
(b) Collapse behaviour of a fire engineering designed single-storey cold-formed steel building in severe fires, Thin-Walled Structures 142, 340-357.
(c) The collapse behaviour of cold-formed steel portal frames at elevated temperatures, RPD Johnston, M Sonebi, JBP Lim, CG Armstrong, AM Wrzesien, ..., Journal of Structural Fire Engineering
(d) Finite-element investigation of cold-formed steel portal frames in fire, RPD Johnston, JBP Lim, HH Lau, Y Xu, M Sonebi, CG Armstrong, CC Mei, Proceedings of the Institution of Civil Engineers-Structures and Buildings.
- The presentation of the results could be improved.
- Figure 5(b), are these dimensions in “mm”, then please mention it in the Figure title. The same comment for Figure 6.
Author Response
We thank the Reviewer for their valuable comments and we list our response below.
Abstract section should be improved considering the following structure: introduction, problem statement, methodology, results, and conclusion.
We adjusted the abstract according to the suggested structure.
The paper is technically interesting; however, the novelty of this paper should be further justified and to establish the contributions to the new body of knowledge (mainly in the abstract and introduction sections).
In the Introduction section, the authors should improve the research background, the review of significant works in the specific study area, the knowledge gap, the problem statement, and the novelty of the research.
We rearranged both the Abstract and the Introduction to acknowledge the Reviewer clearly state the knowledge gap, the problem statement, and the novelty of the research. Particularly:
Knowledge gap.
From the introduction: When data are available, we can perform the thermal compensation by fitting the monitoring data with a model that accounts for temperature effects. By fitting with a probabilistic method, we can also quantify the posterior uncertainty introduced by the thermal compensation and judge whether its magnitude is acceptable or not. However, at the design stage the monitoring system is not yet installed, and no recording is available. So how can we predict what is the error introduced by thermal compensation? And how can we design a proper monitoring system to keep this un-certainty below an acceptable level?
From the abstract: Estimating the uncertainty of the condition state is relatively straightforward a posteriori, i.e., when the monitoring data are available. However, when we design a monitoring system, the monitoring observations are not available yet; therefore, an expected uncertainty must be estimated beforehand.
Problem statement (and method).
From the introduction: In this paper, we wish to answer to these questions, by introducing a logical approach to estimate a pre-posteriori the expected uncertainty of a long-term structural response trend. The formulation we propose accounts for the errors due to temperature compensation, sensors performance, and interpretation model.
From the abstract section: We propose a framework to evaluate the effectiveness of a monitoring system accounting for temperature compensation, which applies to the design process of a tentative structural health monitoring system for civil infrastructures. Particularly, we focus on the condition-state parameters representing the structural long-term response trend, e.g., due to creep and shrinkage effects and the tension losses in prestressed concrete bridges.
Novelty of the research.
From the introduction: It (the formulation) allows estimating the expected un-certainty before monitoring data are available, thus helping the monitoring system de-signers to answer the following design questions: (i) What sensor technology and measurement accuracy are required? (ii) How long the monitoring duration should be to satisfy the target uncertainty? (iii) What is the minimum sampling frequency to satisfy the target uncertainty given a certain monitoring duration?
From the abstract: The result is a simple-to-use equation that estimates in the design phase the expected uncertainty of a long-term response-trend of temperature compensated response measurements. The equation clearly demonstrates how the condition-state uncertainty is affected by measurements and model uncertainties, start date and duration of the monitoring acquisition, and sampling frequency. We validated our approach on a real-life case study: the Colle Isarco viaduct. We verified whether the expected uncertainty estimated a pre-posteriori with our approach is consistent with the real uncertainty estimated a posteriori based on the monitoring data.
Introduction is not organized well. The reviewer cannot readily see the significance of this study compared to the previous works. The literature review should be extended to more recently published works available in the literature.
As per Reviewer’s request, we broadened the literature review in the introduction section, including the most significant works on that matter regarding the effect of environmental temperature on civil structural health monitoring. These include:
[17] H. Sohn, “Effects of environmental and operational variability on structural health monitoring,” Phil. Trans. R. Soc. A, vol. 365, p. 539–560, 2007.
[27] H. Abdel-Jaber e B. Glisic, «Monitoring of prestressing forces in prestressed concrete structures—An overview,» Structural Control and Health Monitoring, vol. 26, n. 8, pp. 1-27, 2019.
[28] H. A. Jaber e B. Glisic, «Monitoring of long-term prestress losses in prestressed concrete structures using fiber optic sensors,» Structural Health Monitoring, vol. 18, n. 1, pp. 254-269, 2019.
[34] E. Cross, K. Koo, J. Brownjohn and K. Worden, “Long-term monitoring and data analysis of the Tamar Bridge,” Mechanical Systems and Signal Processing, vol. 35, pp. 16-34, 2013.
[35] C. Rodrigues, C. Félix, A. Lage and J. Figueiras, “Development of a long-term monitoring system based on FBG sensors applied to concrete bridges,” Engineering Structures, vol. 32, p. 1993–2002, 2010.
[36] A. Kita, N. Cavalagli and F. Ubertini, “Temperature effects on static and dynamic behavior of Consoli Palace in Gubbio, Italy,” Mechanical Systems and Signal Processing, vol. 120, pp. 180-202, 2019.
We cannot avoid to notice, that, among the rest, this Reviewer suggested citing a number of works on fire performance of steel frame structure, a topic which is unrelated to this manuscript. We remind that our paper deals with environmental (not fire) thermal compensation of structural health monitoring data. After careful reading the papers recommended by this Reviewer, we can only conclude that none of the method included in them has connection with the present research work.
We ignore the reason that prompted the Reviewer to recommend citing these particular papers. Whatever the reason is, we iterate these are all unrelated to our manuscript, and we felt inappropriate to include any of them in our state of the art.
More precisely:
(a) Rahman, M., Lim, J. B., Xu, Y., Hamilton, R., Comlekci, T., & Pritchard, D. (2013). Effect of column base strength on steel portal frames in fire. Proceedings of the Institution of Civil Engineers-Structures and Buildings, 166(4), 197-216.
This paper focuses on a finite-element model of a steel portal frame building in fire and its use to assess the adequacy of the Steel Construction Institute design method. Our manuscript focuses on environmental temperature compensation of structural health monitoring data and we do not find the suggested paper relevant for our literature review.
(b) Roy, K., Lim, J. B., Lau, H. H., Yong, P. M., Clifton, G. C., Johnston, R. P., ... & Mei, C. C. (2019). Collapse behaviour of a fire engineering designed single-storey cold-formed steel building in severe fires. Thin-Walled Structures, 142, 340-357.
This one describes a full-scale natural fire test to investigate the collapse behaviour of a single storey cold-formed steel (CFS) building, designed to behave in a specified way in a severe fire, with roof venting and partial wall collapse. It does not bring any relevant contribution to our state-of-the-art review, since we do not study how steel structure behave in a severe fire.
(c) Johnston, R. P., Sonebi, M., Lim, J. B., Armstrong, C. G., Wrzesien, A. M., Abdelal, G., & Hu, Y. (2015). The collapse behaviour of cold-formed steel portal frames at elevated temperatures. Journal of Structural Fire Engineering.
This describes the results of non-linear elasto-plastic implicit dynamic finite element analyses that are used to predict the collapse behaviour of cold-formed steel portal frames at elevated temperatures. Again not related to our research topic.
(d) Finite-element investigation of cold-formed steel portal frames in fire, RPD Johnston, JBP Lim, HH Lau, Y Xu, M Sonebi, CG Armstrong, CC Mei, Proceedings of the Institution of Civil Engineers-Structures and Buildings.
This presents the results of a full-scale site fire test performed on a cold-formed steel portal frame building with semi-rigid joints. The purpose of the study is to establish a performance-based approach for the design of such structures in fire boundary conditions. Our paper focused on monitoring systems design not structural design and there is no connection to our literature review.
The presentation of the results could be improved.
We modified most figures making them more readable and clearer thanks to the Reviewers’ suggestions.
Figure 5(b), are these dimensions in “mm”, then please mention it in the Figure title. The same comment for Figure 6.
Thanks for noticing this, we added the correct dimension, “meters”, in Figures 5 and 6.
Round 2
Reviewer 2 Report
How would someone differentiate curves shown in Fig. 5 if printed in black and white? The same comment for Figs. 4 and 7.
Section 5.3, :I wouldn't normally write a paper in active voice and use terms like "we validate"....it would be better to use passive voice, such as....". The proposed approach was validated....". The whole paper is currently written in active voice which is not very common to use, particularly for writing research articles.
The English language should be significantly improved throughout the paper, there are grammatical mistakes, missing links between paragraphs, some of the sentences are bland. The writing needs to be significantly improved before the next version can be reviewed.
Author Response
We thank the Reviewer for their valuable comments and we list our response below.
- How would someone differentiate curves shown in Fig. 5 if printed in black and white? The same comment for Figs. 4 and 7.
We understand the Reviewer point, but Figures are deliberately in color as recommended by the Instructions for Authors of this Journal: “Authors are encouraged to prepare figures and schemes in color (RGB at 8-bit per channel). There is no additional cost for publishing full color graphics.” Infrastructures | Instructions for Authors (mdpi.com)
- Section 5.3, :I wouldn't normally write a paper in active voice and use terms like "we validate"....it would be better to use passive voice, such as....". The proposed approach was validated....". The whole paper is currently written in active voice which is not very common to use, particularly for writing research articles.
We changed most of the sentences according to the Reviewer request. However, we kept the active voice in the sentences where the subjects are the Authors. Indeed, our deliberate policy is to use first person in scientific writing, and we no longer align with the outdated academic writing style recommendation to refer to the authors in third person. We believe that in modern scientific communication the active form is preferable in most of cases, because it is more direct, clearer and simpler to understand. Regardless our personal opinion, we notice that most modern, and not so modern, style manuals strongly recommend using the first person rather than the passive voice. Even ASCE, traditionally one of the most radical advocate of third person, has changed its position and since 2000 their style guide advises: "Wherever possible, use active verbs... and ...the use of “I” and “we” is preferable to awkward constructions such as “the authors” or “this researcher.”
- The English language should be significantly improved throughout the paper, there are grammatical mistakes, missing links between paragraphs, some of the sentences are bland. The writing needs to be significantly improved before the next version can be reviewed.
As per the Reviewer request, we had the paper proofread by a professional English editing service. We uploaded the revision certificate for the Editor.